# Diversification and extinction of Hemiptera in deep time
Mathieu Boderau [1] ✉, André Nel [1,3] & Corentin Jouault [2,3]

Untangling the patterns and drivers behind the diversification and extinction of highly diversified lineages remains a challenge in evolutionary biology. While insect diversification has been widely studied through the "Big Four" insect orders (Coleoptera, Hymenoptera, Lepidoptera and Diptera), the fifth most diverse order, Hemiptera, has often been overlooked. Hemiptera exhibit a rich fossil record and are highly diverse in present-day ecosystems, with many lineages closely associated to their host plants, making them a crucial group for studying how past ecological shifts—such as mass extinctions and floral turnovers—have influenced insect diversification. This study leverages birth-death models in a Bayesian framework and the fossil record of Hemiptera to estimate their past diversity dynamics. Our results reveal that global changes in flora over time significantly shaped the evolutionary trajectories of Hemiptera. Two major faunal turnovers particularly influenced Hemiptera diversification: (i) the aftermath of the Permo-Triassic mass extinction and (ii) the Angiosperm Terrestrial Revolution. Our analyses suggest that diversification of Hemiptera clades was driven by floristic shifts combined with competitive pressures from overlapping ecological niches. Leveraging the extensive fossil record of Hemiptera allowed us to refine our understanding of diversification patterns across major hemipteran lineages.

Changes in biodiversity punctuate the evolution of life on Earth through time, reflecting fundamental mechanisms of species diversification and extinction[1–5]. While many studies have attempted to decipher the timing and tempo of lineage diversifications and declines using time-calibrated phylogenetic trees[6–8], recent research has highlighted several limitations associated with these approaches[9–11]. These concerns are significant and have far-reaching implications, as they affect all widely used methods for studying diversification. However, recent evaluations of these issues indicated that non-identifiability does not render current methods unusable[12–15]. Crucially, there has been no formal demonstration that Bayesian, fossil-based, process-driven approaches to diversification suffer from identifiability issues. Therefore, approaches using the fossil record offer valuable alternatives for studying lineage diversification (Fig. 1A–I). The fossil record provides empirical data to estimate these dynamics, and valuable insights into extinction events, recovery phases and faunal turnovers, and affords a window into the interplay between shifts in diversity and past environmental changes[2,16–25]. New models have been developed, being able to jointly estimate speciation, extinction and preservation rates, as well as their temporal variations, using fossil occurrence data while effectively controlling for heterogeneity in the fossil record[26,27]. Some of these models have proven to

be robust to a range of potential biases[27,28] (e.g. singletons exceeding 30%, violations of sampling assumptions and variable preservation rates). Although these approaches have many advantages, they remain rarely used in the study of insect evolutionary dynamics[23,24,29].

Among the most diverse insect lineages with a significant fossil record, Hemiptera rank as the fifth most speciose group[30]. This order comprises over 107,000 extant species[31]—encompassing aphids, scale insects, tree-hoppers, leafhoppers, planthoppers and both predatory and plant-feeding bugs (Fig. 1J). The Hemiptera are also diverse in deep time, with over 3350 extinct species (PBDB; accessed on 06/20/2024) with more than 96% classified into its major extant sub-orders. No order of insects includes more extant and extinct families than the Hemiptera[32]. Their oldest fossils date back to the Late Carboniferous (320 million years ago, Ma), with the Protoprosbolidae and Aviorrhynchidae lineages[33,34]. Recent divergence time analyses suggested that Hemiptera may have originated even earlier, in the Late Devonian-Early Carboniferous[35,36].

Phytophagy is thought to have contributed to the evolutionary success of the Hemiptera by allowing them to switch plant hosts during major environmental upheavals and floral turnovers[37]. These multiple shifts of food diet were promoted by the convergent and repeated acquisition of different

[1]Institut de Systématique, Évolution, Biodiversité (UMR 7205) Muséum National d'Histoire Naturelle, CNRS, Sorbonne Université, EPHE-PSL, Université des Antilles, Paris, France. [2]Oxford University Museum of Natural History, University of Oxford, Oxford, UK. [3]These authors jointly supervised this work: André Nel, Corentin Jouault. ✉e-mail: mathieuboderau@gmail.com

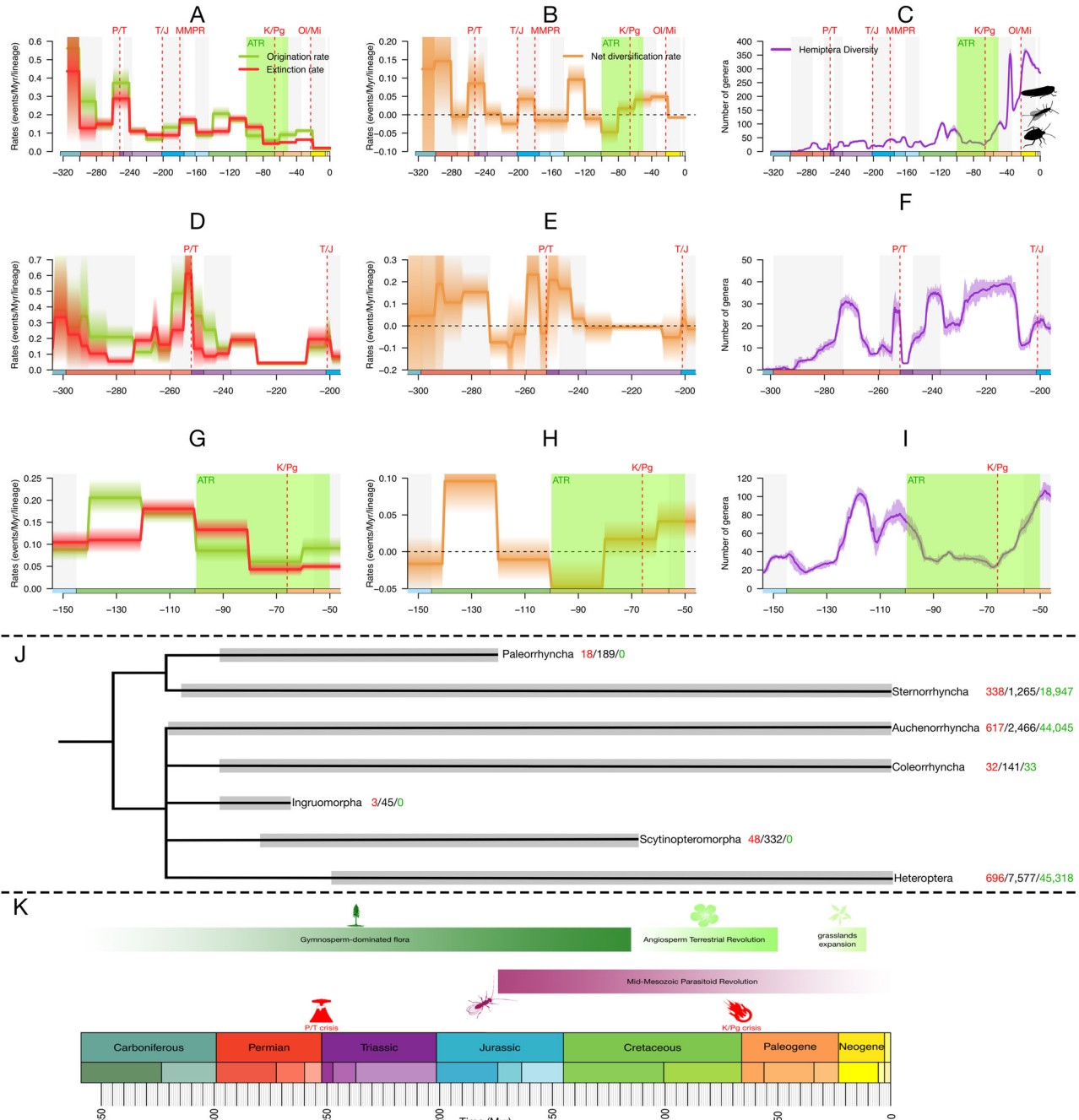

**Fig. 1 | Diversification and diversity dynamics of Hemiptera (genus-level analysis). A**, **D**, **G** Bayesian estimates of origination (green) and extinction (red) rates for Hemiptera. **B**, **E**, **H** Net diversification rate (origination minus extinction rates) for Hemiptera. **C**, **F**, **I** Diversity through time (number of genera) of Hemiptera. **J** Relationships between major extinct and extant Hemiptera lineages (adapted from Johnson et al.[36] and Szwedo[32]), showing the number of genera included in our analyses (in red), the number of occurrences in our datasets (in black) and the number of extant species (based on Streito and Germain[150]), the lifespan of each lineage indicated by the grey bars. **K** Chronostratigraphic scale since Carboniferous with the major paleo-events. BDCS analyses were performed under 20-Myr time bins (**A**, **B**, **C**, **G**, **H**, **I**) and 5-Myr time bins (**D**, **E**, **F**). Solid lines indicate means posterior rates while shaded areas show 95% CI. ATR Angiosperm Terrestrial Revolution, K-Pg Cretaceous-Paleogene Event, MMPR Starting of the Mid-Mesozoic Parasitoid Revolution, Ol/Mi Oligocene–Miocene global cooling and drying, P/T Permian-Triassic Event, T/J Triassic–Jurassic Event. Time-unit is millions of years (Myr). Colours of geological periods in chronostratigraphic scale following International Chronostratigraphic Chart (ICS, v2023/09). Insect and plant silhouettes from http://phylopic.org/ and licences at https://creativecommons.org/publicdomain/zero/1.0/.

endosymbionts and by the evolution from the 'ancestral' piercing-sucking mouthparts shared with other acercarian lineages (Permopsocida and Thysanoptera) to rostrum with a multi-segmented sheet-like labium covering the mandibular and maxillary stylets[38–40]. This rostrum is considered a major evolutionary innovation that contributed to shaping the evolutionary history of this clade, notably allowing a switch from pollen/spore feeding to plant-fluid feeding[41] (phloem: Auchenorrhyncha and Sternorrhyncha; xylem: some cicadomorphan groups, such as cicadas and spittlebugs (Cercopoidea)) and diversification into novel ecological niches[37,42].

The shift from phytophagy to predation is also regarded as a factor driving the diversification of this group[37]. The most diverse Hemiptera lineage, Heteroptera (true bugs), which encompasses half of all extant

species in the order, originated synchronously with a shift from phytophagy to predation[43]. Interestingly, many true bugs (Pentatomomorpha and Cimicomorpha) later convergently shifted back to phytophagy, exploiting a variety of plant-derived resources (fruits or seeds[44]). These shifts are hypothesised to be linked to the diversification of Angiosperms, potentially playing a significant role in the evolutionary success of Heteroptera[45]. Because of their feeding habits (either phytophagous or predators), many bugs are also regarded as pests, vectors of plant pathogens and diseases, and even vectors of human diseases, as hematophagy appeared convergently within the Cimicomorpha in assassin bugs and bed bugs[46–49].

The diversification of Hemiptera and other phytophagous insects during the Mesozoic likely fuelled one of the most transformative events of this era: the diversification of parasitoid insects[50]. This event, known as the Mid-Mesozoic Parasitoid Revolution (MMPR[50]), began over 180 million years ago. The MMPR is characterised by a shift from bottom-to-top regulation of terrestrial food webs, dominated by less efficient clades of predators, to top-to-bottom control by more efficient parasitoid clades. However, this rise in parasitoid diversity may have exerted selective pressures that negatively influenced the diversification of Hemiptera, an impact that remains to be quantified.

Three major paleo-events are suggested to have significantly influenced the evolutionary trajectory of Hemiptera (Fig. 1K). The Permo-Triassic (P–T) mass extinction caused a turnover in hemipteran fauna between the Paleozoic and Mesozoic, as it did for many other insect groups[23,51]. A second significant turnover occurred during the Angiosperm Terrestrial Revolution (ATR between 100 and 50 Ma[52]), leading to the extinction of many specialised and relict hemipteran groups[53]. Finally, the global cooling and drying during the Oligocene–Miocene spurred the expansion of grasslands —favouring the diversification of certain plant lineages and a change from C4 to C3 metabolic pathways (e.g. orchids[54])—which likely contributed to the emergence of new specialised hemipteran faunas composed by many cicadomorphan families, aphids and Delphacidae[32,55,56]. Although these hypotheses have been proposed, they have yet to be rigorously tested across the entire Hemiptera clade.

In this study, we aim to test some of these key evolutionary hypotheses about Hemiptera diversification using their extensive fossil record to clarify their diversification dynamics and identify the potential environmental factors that may have influenced the wane and wax of their constitutive lineages. We have compiled and analysed a dataset of over 11,840 fossil occurrences, for 244 families and 1794 genera, from deposits worldwide. We estimate origination (speciation above the species level) and extinction rates and their variations over time, relying on a Bayesian framework and birth-death models that account for preservation biases and uncertainties of the fossil record[27,28]. These estimates allow us to propose a reconstruction of the evolutionary history of the order Hemiptera and its constitutive lineages. Additionally, we investigate how changes in the abundance of plant lineages over geological time have influenced the origination and extinction rates of Hemiptera. Our results reveal correlations between the diversity dynamics of Hemiptera and the evolution of plant relative abundance in past ecosystems, with clade-specific origination and extinction rates showing varied responses to environmental factors.

## Results
### Global diversification pattern of Hemiptera
Our analyses indicate that Hemiptera evolutionary history is punctuated by a succession of diversification and extinction periods of varying intensities and durations. Hemiptera diversified from the late Carboniferous to the middle Permian (Fig. 1A, B), with only a short episode of decline during the middle-late Permian related to a drop in the origination rate (Fig. 1A). They underwent significant diversification from the late Permian to the Middle Triassic (origination rate increasing by ≈2.5 fold the origination rate through middle Permian), this increase impacted the late Paleozoic lineages in Auchenorrhyncha and Sternorrhyncha (Fig. 2A–D). This result is corroborated by our different analyses (genus and family levels; Figs. 1A and 2A, D; Supplementary Figs. 4–7). During the same period, the extinction

rates for Hemiptera (genus level) increased, particularly for the Auchenorrhyncha with the extinction rate being almost equal to the origination rate (Fig. 2A).

Our analyses, focusing on the P–T and using short time bins (genus level, 5 Ma; Fig. 1D–F), allow us to better frame the timing of the extinction events, and show that the extinction rate reached a maximum at the boundary of these two periods (from 0.25 at the beginning of the late Permian to 0.6 at the P–T boundary; Fig. 1D). The extinction rate decreased sharply after that boundary, a result supported by additional analyses (at the genus level) with a different model for Hemiptera, Sternorrhyncha and Auchenorrhyncha (see Supplementary Figs. 1 and 2C, E).

Hemiptera underwent a second episode of decline during the Late Triassic (Fig. 1B, C), with the origination rate dropping from 0.11 to 0.06 in 20 Ma (Fig. 1A). This decline mainly affected the Auchenorrhyncha, with the net diversification rate becoming negative at the end of the Triassic (Fig. 2B) due to a decrease in the origination rate, from 0.3 at the beginning of the Triassic to 0.05 at the Triassic–Jurassic boundary.

The next episode of diversification for Hemiptera occurred during the Early Jurassic (Fig. 1A), with a significant increase in the origination rate (from 0.06 to 0.13; Fig. 1A). This pattern is recorded within all hemipteran major lineages at the genus level (Fig. 2B, E, H), with notably the strongest increase in the origination rate of Heteroptera (reaching 0.7; Fig. 2G).

At the genus level, Hemiptera declined from the Middle Jurassic to the Late Jurassic (Fig. 1B). However, the impact of this decline is heterogeneous between Hemipteran major lineages (Fig. 2B, E, H). Auchenorrhyncha diversification, at the genus level, started to slow down and declined during the Middle and the Late Jurassic with a net diversification rate decreasing more than tenfold relative to the Early Jurassic median. A similar pattern is found for the Sternorrhyncha (at the genus level). However, the diversity dynamics of the Heteroptera contrasts with the decline of the two other groups as their extinction is mainly clustered during the Middle Jurassic. Interestingly, Heteroptera diversification sharply increases during the Late Jurassic and Early Cretaceous.

The Early Cretaceous marked another period of diversification for the Hemiptera, with the second-highest peak of net diversification rate recorded in genus-level analyses (Fig. 1C). This peak is consistent across all analyses and under both the birth-death model with constrained shifts (BDCS) and reversible-jump Markov Chain Monte Carlo (RJMCMC) models (Figs. 1H and 2B, E, H; Supplementary Figs. 4–7). Auchenorrhyncha and Heteroptera diversification are key components of this peak, their origination rates being more than two-fold higher than their extinction rates (Fig. 2A, G). At the family level, the peak of diversification is mainly driven by an increase in the Sternorrhyncha origination rate, about threefold higher than the extinction rate (Supplementary Fig. 2).

The last remarkable period for Hemiptera is the ATR. During the early stages of this event, we recorded a period of decline with the origination rate dropping more than the extinction rate, leading to a drop in the net diversification rate (from 0 to −0.05; Fig. 1G). This result is consistent across all datasets and analyses at the genus level and under both the BDCS and RJMCMC models. Interestingly, during the second half of the ATR, the extinction rate significantly decreased for Auchenorrhyncha, Heteroptera and Sternorrhyncha (Fig. 2A, D, G) whereas the origination rate is relatively constant for Heteroptera and increased for the Auchenorrhyncha. The Sternorrhyncha declined through the early stages of the ATR but started to diversify again during the latest stages of this period, with a major increase in the origination rate through the Paleocene and the Eocene periods (Fig. 2D–F).

The Oligocene–Miocene transition is marked by a decrease in both origination and extinction rates in all datasets and analyses at the genus level (Figs. 1A and 2A, D, G).

Our lineage-though-time (LTT) analysis shows that the number of Hemiptera genera fluctuated continuously between the origin of the clade and the end of the Cretaceous (Fig. 1C). Through the Cretaceous, two peaks of diversity are found at 117 and 102 Ma with, respectively, over 100 and 80 genera (Fig. 1C). Hemiptera diversity declined during the

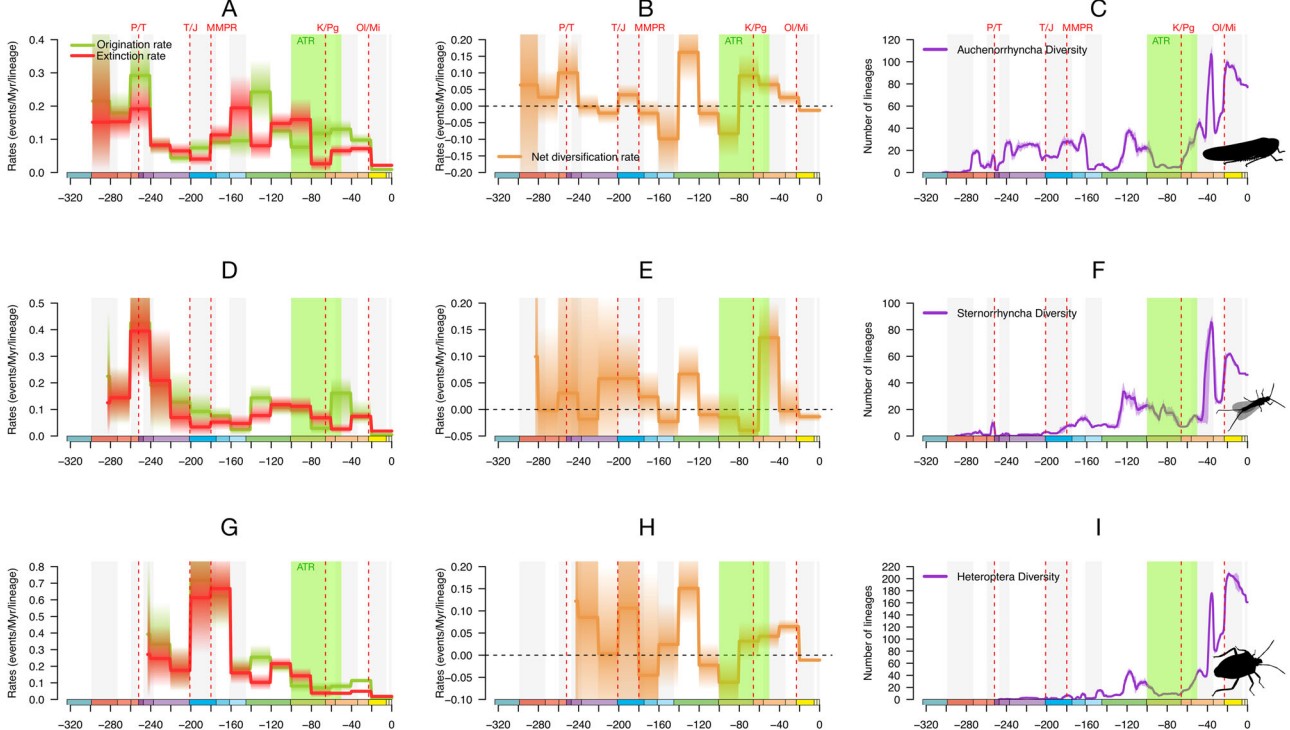

**Fig. 2 | Asynchronous diversification and diversity dynamics within Hemiptera (genus-level analysis). A, D, G** Bayesian estimates of origination (green) and extinction (red) rates for Hemiptera major clades. **B, E, H** Net diversification rate (origination minus extinction rates) for Hemiptera major clades. **C, F, I** Diversity through time (number of genera) of Hemiptera major clades. Solid lines indicate means posterior rates while shaded areas show 95% CI. ATR Angiosperm Terrestrial Revolution, K-Pg Cretaceous-Paleogene Event, MMPR Starting of the Mid- Mesozoic Parasitoid Revolution, Ol/Mi Oligocene–Miocene global cooling and drying, P/T Permian-Triassic Event, T/J Triassic–Jurassic Event. Time-unit is millions of years (Myr). Colours of geological periods in chronostratigraphic scale following International Chronostratigraphic Chart (ICS, v2023/09). Insect silhouettes from http://phylopic.org/ and licences at https://creativecommons.org/publicdomain/zero/1.0/.

early stages of the ATR and sharply increased during the late ATR. Through the Cenozoic, two peaks are found at 36 Ma (slightly before the Eocene–Oligocene transition) with 354 genera, and at 17 Ma with 364 (early Miocene). These two diversity peaks were recorded in Auchenorrhyncha, Sternorrhyncha and Heteroptera (Fig. 2C, F, H). The Eocene–Oligocene transition corresponds to the highest diversity recorded for Auchenorhycnha and Sternorrhyncha, with, respectively, more than 100 and 80 genera (Fig. 2C, F), whereas the highest diversity recorded for Heteroptera is at the Oligocene–Miocene boundary with more than 200 genera (Fig. 2H). Our LTT plot shows a decrease in genus number after the Eocene peak of diversity, likely corresponding to the Baltic amber, and between the Miocene to the present days (reaching over 285 genera). At the family level (Supplementary Fig. 3D), the pattern is strikingly different, with a nearly linear increase in the number of families from the Carboniferous to the mid-Cretaceous (reaching a peak of diversity of 88 families at 96 Ma), and explained by the accumulation of extinct and extant families. Then the number of families decreased through the ATR following a pattern similar to genus-level analysis, before sublinearly increasing between the late Cretaceous to the present days, with about 105 families recorded during the Miocene and the rest of the Cenozoic.

**Correlations with biotic and abiotic variables**
The results of our multivariate birth-death (MBD) analyses are summarised in Supplementary Tables 1–8. At the genus level (Fig. 3), Hemiptera origination is positively correlated with Angiosperm diversification ($\omega \approx 0.78$, $G \approx 1.86$), and fluctuations in the diversity of spore plants ($\omega \approx 0.88$, $G \approx 4.37$) and non-Polypodiales ferns ($\omega \approx 0.81$, $G \approx 3.62$). However, an intra-order diversity-dependence effect is also found, with the origination of Hemiptera being negatively correlated with the changes in genus diversity

through time ($\omega \approx 0.53$, $G \approx -0.85$). Hemiptera extinction is positively correlated only with fluctuations in the diversity of non-Polypodiales ferns ($\omega \approx 0.8$, $G \approx 3.51$).

Auchenorryncha origination (Fig. 3) is positively correlated with the diversification of angiosperms ($\omega \approx 0.79$, $G \approx 1.81$) and fluctuations in abundance of non-Polypodiales ferns ($\omega \approx 0.88$, $G \approx 4.84$) in past environments. Moreover, Auchenorryncha origination shows a negative correlation with the decline of Gymnosperms ($\omega \approx 0.8$, $G \approx 3.51$) and a diversity-dependence effect ($\omega \approx 0.68$, $G \approx -1.21$). The extinction of Auchenorrhyncha is positively correlated with fluctuations in the diversity of non-Polypodiales ferns ($\omega \approx 0.85$, $G \approx 4.23$) and negatively correlated with the decline of Gymnosperms ($\omega \approx 0.61$, $G \approx -1.56$).

The origination of Sternorrhyncha (Fig. 3) is positively correlated with the diversification of Angiosperms ($\omega \approx 0.92$, $G \approx 3.84$), while their extinction is positively correlated with fluctuations in abundance of spore plants ($\omega \approx 0.98$, $G \approx 11.2745$).

A diversity-dependence effect is found for Heteroptera, with their origination being negatively correlated with changes in the diversity of heteropteran genera through time ($\omega \approx 0.68$, $G \approx -1.41$).

At the family level (Fig. 3), our results support a significant positive correlation between Hemiptera origination and the fluctuations in the abundance of spore plants ($\omega \approx 0.92$, $G \approx 5.7$). In contrast, Hemipteran extinction rates show negative correlations with fluctuations in Angiosperm abundance ($\omega \approx 0.92$, $G \approx -3.54$) and Gymnosperm abundance ($\omega \approx -0.84$, $G \approx -3.39$), while extinction is positively correlated with changes in overall family diversity over time ($\omega \approx 0.62$, $G \approx -0.93$).

The extinction rates of Sternorrhyncha, show a positive correlation with fluctuations in family diversity over time ($\omega \approx 0.92$, $G \approx 3.75$). In contrast, Heteroptera origination rates are negatively correlated with changes in heteropteran genus diversity over time ($\omega \approx 0.94$, $G \approx -4.35$).

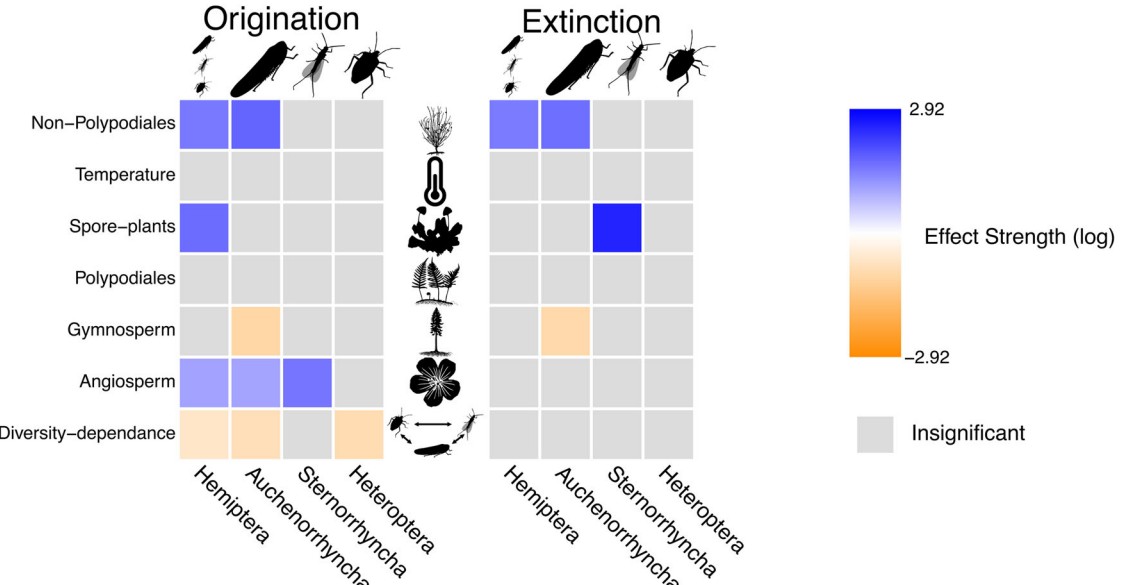

**Fig. 3 | Bayesian inferences of correlation parameters on origination (left panel) and extinction (right panel) for Hemiptera (genus- and family-level analyses) with the set of abiotic and biotic variables: Clade Diversity-dependence; Angiosperms; Gymnosperms, Polypodiales ferns, Spore plants, Temperature and Polypodiales ferns.** The strength effect corresponds to the median value of the correlation parameter to the origination/extinction rate. The effect of a variable is assessed as significant when the shrinkage weight ($\omega$) is superior to 0.5 and 95% HPD does not encompass 0 (See Supplementary Tables). Insect and plant silhouettes from http://phylopic.org/ and licences at https://creativecommons.org/publicdomain/zero/1.0/.

## Discussion

### Early diversification of Hemiptera and Paleozoic–Mesozoic turnover

The oldest hemipterans are described from the Late Carboniferous[33,34], suggesting an even older origin for the clade. Hemiptera diversified during the Permian with the extinct Prosorrhyncha (Ingruomorpha)—this clade is related to Cicadomorpha[32]—and their high morphological disparity. The oldest uncontroversial cicadomorphans, 'coleorrhynchans' and fulgoromorphans are also recorded from Permian deposits suggesting that these lineages may have co-habited in Permian environments[57–60].

The P–T mass extinction triggered a major turnover of Hemiptera lineages[23,61], and we recorded a significant impact of this event, characterised by the decline of Hemiptera, across all our analyses (5 Ma or stage-time bins under both the BDCS and RJMCMC models). This decline is linked with major increases in extinction rates in hemipteran lineages (Fig. 1D). The P–T extinction event coincides with the biggest turnover in the evolutionary history of Hemiptera, with numerous Paleozoic groups wiped out—with the exceptions of some relict lineages (Palaeontinoidea, Progonocimicoidea or Scytinopteromorpha)—or being replaced by modern lineages of Hemiptera that diversified throughout the Mesozoic. This extinction is directly evidenced in the fossil record because it corresponds to the transition from the Paleozoic Insecta Fauna (e.g. extinction of Palaeodictyopteroidea) to Mesozoic Insecta Fauna (rise and diversification of the crown Diptera and Hymenoptera and diversification of Coleoptera and Hemiptera)[51,62,63].

The P–T extinction event is characterised by a series of drastic changes in aquatic and terrestrial ecosystems triggered by massive volcanic eruptions occurring in Siberia[64–67], which caused global warming, combined with acid rains due to volcanic gas emissions, and led to a loss of vegetation cover and decline of plants[68]. Therefore, it is not surprising that hemipteran lineages of this period declined, being closely associated with plants for feeding or 'nesting'.

The diversification of the clade resumes rapidly during the Triassic, with the rise and diversification of Heteroptera[37,69]. The diversification of aquatic true bugs—the oldest known fossil true bugs are aquatic and belong to the Nepomorpha clade—could have been made possible by major changes in freshwater ecosystems and the co-diversification with aquatic larvae of other insect clades[22,29] constituting a wide array of prey for these insects[70] and by the ecological niches left unoccupied after the P–T mass extinction[71]. The early Mesozoic diversification of Hemiptera could be linked with ecological niches left vacant by the Paleozoic group of Hemiptera that went extinct due to environmental changes, but also by the extinction of potentially competing groups like the superorder Palaeodictyopteroidea that also possessed rostrum, probably linked to the same type of interactions with plants[49,72,73].

The Triassic–Jurassic extinction event had a reduced impact on Hemiptera, aligning with results from previous studies on insects[20,62,74]. This crisis, driven by volcanic activity at the end of the Triassic, slightly impacted the plant communities[75,76]. Floral changes were not as severe as those during the end-Permian extinctions, and do not correspond with major extinction events for plant lineages[77–79]. Hemiptera's phytophagous lineages were highly diversified during the Triassic[37], and the absence of major changes in their host plants may account for the observed patterns in our results (i.e. no major floral turnover equals no major extinction in Hemiptera).

Throughout the Jurassic, Hemiptera diversified (Fig. 1B, C), especially the Auchenorrhyncha and Heteroptera lineages (Fig. 2B, C, E, F). During this period, the diversity of these two groups increased simultaneously with their morphological disparity[37]. While Fulgoromorpha originated in the Permian[59], their major radiation occurred around 180 Ma, with the emergence of families such as 'Fulgoridiidae', or Qiyangiricaniidae[80,81]. A similar pattern of diversification is observed in other Auchenorrhyncha lineages, like the Cicadomorpha (e.g. Dysmorphoptiloidea, the Palaeontinidae, or the Hylicellidae). Interestingly, the earliest records of extant superfamilies have also been described from Jurassic deposits (e.g. Cercopoidea (Procercopidae and Sinoalidae) or Cicadelloidea (Archijassidae))[82].

During the Jurassic, ferns and gymnosperms diversified[52,76], which likely contributed to the success of Hemiptera lineages. Notably, the Jurassic period likely represents one of the richest intervals for the Hemiptera fossil record, with particularly well-studied deposits from the Early Jurassic, such as those in the Daohugou and Karabastau Formations. The Jurassic was a crucial period for Heteroptera as well, with the first record of Cimicomorpha, including the Miridae family, which is now the most diversified phytophagous family of true bugs, comprising over 13,000 extant species[83,84]. Moreover, the earliest record of semi-aquatic bugs (Gerromorpha) was also documented during this period[85]. The Jurassic represents the last well-

documented period for Coleorrhyncha, with numerous fossils of the extinct families Progonocimicidae or Karabasiidae[86]. After the Jurassic, only one occurrence of Coleorrhyncha is known from the Cretaceous[87]. Rapid climate changes occurred throughout the Jurassic[88], a period cooler than the Triassic and Cretaceous, but still warmer than today, which likely promoted the development of diverse terrestrial flora and fauna[74].

The diversification dynamics of Hemiptera during this period may have been also shaped by the MMPR, an event that reorganised terrestrial food webs[50]. Although Hemiptera do not include parasitoid species, many hemipteran lineages serve as hosts for parasitoid lineages[50]. Given the time required for parasitoids to establish and impact host populations, a delay is expected between their emergence and the observable effects on Hemiptera. We hypothesise that the heightened extinction rates among Hemiptera during the latter part of the Jurassic may have resulted from increased selective pressures by parasitoids, potentially driving certain lineages to extinction and by the absence of Lagerstätte.

Our analyses reveal an increase in the extinction rate for Auchenorrhyncha and Heteroptera—the two groups frequently targeted by parasitoids—at the onset of the MMPR (Fig. 2A, G). This pattern suggests that the MMPR, and the diversification of parasitoid lineages, may have negatively impacted Hemiptera diversification, particularly during the Jurassic.

Our results suggest that the early evolutionary history of Hemiptera was characterised by a significant turnover between Paleozoic Insect fauna and 'modern' Mesozoic Insect fauna.

## The mid-Cretaceous biotic reorganisation of the biosphere as a linchpin for the diversification of hemipteran lineages

The mid-Cretaceous biotic reorganisation of the biosphere is, inter alia, related to the extinction of Gymnosperms, the rise in dominance of Angiosperms and climate changes (ATR[89]). This is a major step in the evolutionary history of Insects, including Hemiptera[25] (Figs. 1B, 2F and 3B, E, H). We detected significant changes in origination and extinction rates and diversity during this period, with first a period of extinction followed by a period of diversification during the ATR (Fig. 3). Interestingly, our study of the Hemiptera fossil record indicates that this pattern corresponds to the extinction of specialised Mesozoic and relic Paleozoic taxa and is followed by the diversification of modern lineages (especially phytophagous ones). This trend is well-documented within the fossil record for Cicadomorpha with the last known occurrences (extinctions) of stem lineages (e.g. Palaeontinoidea, Hylicelloidea, Dysmorphoptiloidea) and the diversification of crown lineages (e.g. Cicadellidae, Myerslopiidae and the first singing cicadas (Cicadidae)[90,91].

For the Fulgoromorpha, no significant clade vanished during the ATR but many families were first recorded and started to diversify during this period[92]. Within the Sternorrhyncha, the Protopsyllidioidea, a group with many species described from late Paleozoic and Jurassic deposits, is lastly recorded during the early Late Cretaceous[93].

Many extant families of aphids (scale insects) were first documented during the Late Cretaceous. Synchronously, short-living highly specialised groups such as the Dinglomorpha were also present in ecosystems[94]. The Late Cretaceous is also crucial for the diversification of Heteroptera, likely for phytophagous taxa, and previous analyses estimated their diversification to start shortly after the radiation of Angiosperms[45].

In our analyses, the ATR resulted in a Heteroptera extinction, which impacted mostly Jurassic genera, followed by an increase in origination corresponding to the radiation of phytophagous true bugs lineages (Fig. 2G). As for sternorrhynchan lineages, some short-living putatively highly specialised families of Heteroptera are recorded during the mid-Cretaceous, and most of them belong to the Cimicomorpha or the Pentatomomorpha[95–97].

Overall, the mid-Cretaceous appears to be a turning point in Hemiptera evolutionary history with the extinction of both relict Paleozoic taxa and specialised Mesozoic taxa and the radiation of modern hemipteran lineages (Fig. 3). This significant taxonomic change is probably linked to the ATR with the extinction or decline of taxa associated with non-flowering plants and the diversification of lineages associated with Angiosperms[45,94].

## Dynamics of hemipteran lineages were driven by shifts of plant dominances and linked with ecological niche transitions

Our results indicate that the evolutionary history of the Hemiptera has been punctuated by two major extinction events (i.e. the P–T mass extinction, and during the ATR) resulting in major faunal turnovers. MBD analyses indicate that Hemiptera origination and extinction rates correlated only with changes in the relative abundance of plant lineages suggesting that hemipteran diversification was linked to the wane and wax of plant groups (Fig. 3). Our study also extends the quantification of the ATR impact on Hemiptera by analysing data at the genus-level data, whereas previous studies primarily focused on family-level dynamics[25].

Firstly, in our genus-level analyses, we estimated a negative correlation between the origination rate of Hemiptera and the fluctuations in Hemiptera diversity in past environments. This result suggests that as hemipteran diversity increased, origination decreased, emphasising an underlying mechanism of competition or diversity dependence. Phytophagy is widespread within Hemiptera, using the same evolutionary innovation (the rostrum) suggesting the exploitation of the same resources for many lineages, leading to overlapping ecological niches, and unavoidably to competition (Fig. 3). While it might be speculated that competition with other plant-feeding insect lineages, such as some Coleoptera, Lepidoptera or Kalligrammatidae (Neuroptera), could have influenced hemipteran evolution, key distinctions in feeding strategies reduce this possibility. Unlike hemipterans, these groups predominantly feed on leaves, nectar or recessed ovules, whereas hemipterans, with their specialised rostrum, directly access phloem sap. Thus, competition between these groups is likely minimal.

Our analysis also shows a positive correlation between the origination of the Hemiptera and the diversification of Angiosperms (Fig. 3), indicating that the Hemiptera origination likely increased when Angiosperms diversified, and suggesting that many Hemiptera successfully switched from Gymnosperms to flowering plants. This phenomenon (i.e. colonisation of a new plant host) is known to lead to reproductive isolation, favouring speciation[98]. The rise and diversification of Angiosperms opened new ecological opportunities and niches for Hemiptera, and has likely been a major driving force behind the diversification of the clade. The diversification of many other phytophagous insect lineages, such as Coleoptera, Diptera and Lepidoptera, also occurred during the ATR[20,25]. This diversification of phytophagous insects (i.e. prey) may have subsequently fuelled the diversification of predatory Heteroptera. Future research on Hemiptera diversification could explore how the proliferation of prey influenced the evolutionary trajectory of predaceous Heteroptera.

The diversification of hemipteran lineages is not only correlated with the rise of Angiosperms but also with other plant lineages indicating that floral turnovers in deep time could be related to the timing and tempo of Hemiptera diversification. We found a positive correlation between the fluctuations in spore-plant diversity and Hemiptera origination rate. Pollen- and spore feeding has been suggested as ancestral behaviour for Hemiptera[63]. When considering the extinct Permopsocida—sister group of the Hemiptera + Thripida (total group of Thysanoptera, also with similar biologies)—the ancestral feeding behaviour for Hemiptera is likely fungus-, pollen- or spore feeding[42]. It is likely that early lineages within Hemiptera shared the same ecological niches with Permopsocida. The spore-bearing plants diversified throughout the Early Permian and Early Triassic[76] and hemipteran lineages likely waxed and waned synchronously (Fig. 1B, C), ultimately switching to phloem sap feeding.

For Auchenorrhyncha, we estimated interesting negative correlations between origination and extinction and the decline of Gymnosperms in past environments. We found a buffer effect of Gymnosperms against Auchenorrhyncha extinction (Fig. 3), but also that the decline of Gymnosperms in past environments tends to limit Auchenorrhyncha origination (Fig. 3). Interestingly, the Cicadomorpha and Fulgoromorpha were the most speciose Hemiptera lineages during the Triassic–Jurassic, when Gymnosperms diversified. We propose that they capitalise on Gymnosperm diversification, utilising them as food sources, during periods of environmental changes[63,99], which limited their extinction. However, during the ATR, Gymnosperms

declined and some lineages that were associated with them likely collapsed[37], this could also explain the synchronous decrease of their origination rate (Fig. 1A, G). The decline of Gymnosperms could have also affected hosts or food sources of cicadomorphan lineages by favouring intra-clade competition (Fig. 3)—the decrease of available Gymnosperms clustering Cicadomorpha on the same hosts—and contributed to the decrease of the origination rate in Auchenorrhyncha (Fig. 2A).

Our family-level analysis shows that the diversification of flowering plants had a buffering effect on Hemiptera, reducing their extinction (Fig. 3). Similar results were noted in recent studies focusing on the ATR[24,25]. However, while previous studies identified this buffering effect primarily when Angiosperms dominated terrestrial ecosystems (50–0 Ma), our findings reveal a broader, global impact that extends beyond the post-ATR period. Our analysis also indicates that fluctuations in Gymnosperm abundance in past ecosystems played a role in limiting Hemiptera extinction—a pattern not observed in studies focused on the ATR[25]. Hemiptera origination showed a positive correlation with fluctuations in spore-plant abundance, most of which diversified before the ATR. This suggests that the early diversification of spore plants may have played a key role in promoting Hemiptera diversification in deep time. As for the fluctuations in diversity of the Gymnosperms, this pattern was not observed in studies focused on the ATR[25]. Together, these results suggest that spore plants and Gymnosperm effects may have primarily occurred before the ATR, possibly during the Carboniferous to Jurassic. Our results reinforce the idea that shifts in the dominance of major plant lineages over time, particularly Angiosperms and Gymnosperms, strongly influenced Hemiptera's diversification dynamics[99]. Hemipteran lineages that adapted to exploit emerging, dominant plant groups likely gained an evolutionary advantage over others, contributing to the faunal turnovers reflected in our observed diversification and extinction patterns.

Our analyses suggest that changes in the number of hemipteran families promote extinction, highlighting a diversity-dependent effect. While no diversity-dependent effect was detected at the genus level for Sternorrhyncha, we found that an increase in family-level diversity within this group correlates with higher extinction rates. A similar effect was observed in Heteroptera, where a rise in family numbers appears to suppress origination rates, further underscoring the influence of diversity-dependent dynamics within Hemiptera. These results align well with our hypothesis of intra-clade competition in Hemiptera, likely driven by overlapping ecological niches (Fig. 3).

Our different analyses at the genus- and family levels underscore that correlations between environmental changes and origination or extinction rates can vary by taxonomic level. The differences between previous studies and our results may be due to variations in dataset sizes and completeness, as our data were improved through extensive literature mining and curation beyond direct extraction from the Paleobiology Database (see *Fossil occurrences*).

We emphasise that the diversification and extinction of Hemiptera are likely driven by biotic factors, particularly the co-evolution between these insects and their host plants, with which they share close ecological relationships. Fluctuations in the abundance of plant lineages over geological time represent a major challenge for these groups, often forcing shifts in diversification rates and leading to the extinction of highly specialised lineages[94,100–102]. Additionally, competition both within and between Hemipteran groups *plus* with other phytophagous insects likely played a role in shaping Hemiptera evolutionary dynamics over time.

## Limits of the study

We estimated the past diversity dynamics of Hemiptera using a Bayesian inferences and birth-death model framework, based on the fossil record, to estimate their extinction and origination rates and the putative driving factors behind these dynamics. Such approaches involve inherent biases or limitations with the datasets or methods used.

The fossil record of Hemiptera is impressive but with gaps. In the late Carboniferous, crown Hemiptera lineages are supposedly present, but they remain unrecorded[34]. Additionally, the Carboniferous Hemiptera are known from very few localities, and more extensive fieldworks are required to depict their late Paleozoic diversity. Similarly, there are relatively few latest Permian or latest Cretaceous insect assemblages, leading to some uncertainty on the precise 'dates' of extinctions of many Paleozoic or Mesozoic groups. Moreover, some groups seem to be affected by ghost lineages, like the Enicocephalomorpha with their earliest record from Early Cretaceous amber[95] but with a much older origin (probably more than 100 Ma before, during Triassic[36]). The rarity of amber inclusions in Triassic and Jurassic ambers can explain this gap because these very small insects are less likely to fossilise in lacustrine sediments[103]. This limitation cannot be overcome without additional field and taxonomical work. We used PyRate because it can partially alleviate these biases and correctly estimate past dynamics with incomplete taxon samplings[76].

PyRate is a process-based-model approach based on assumptions that may violate real evolutionary processes. PyRate estimates the diversification rates under the assumption that they are homogenous within a clade but may vary over time, despite other analyses demonstrating heterogeneity through the insect tree of life and time[20,104]. To overcome this limitation, within a group as diverse as the Hemiptera, we have generated sub-datasets to account for this heterogeneity. Additionally, all analyses were performed under a model of preservation that is heterogenous across time (accounting for potential Konservat-Lagerstätte effect) and lineages.

Lastly, the systematic framework of Hemiptera is quite robust with the monophyly of many higher taxa well-supported by phylogenetic analyses. However, the monophyly of many extinct taxa is still unresolved and their incorporation in phylogenetic reconstructions remains to be performed. For instance, the monophyly of many Paleozoic groups has never been tested (Paleorrhyncha, Prosorrhyncha and many extinct families). Some poor delimitations also occurred for Mesozoic taxa such as 'Fulgoridiidae'[80]. To prevent any problem with these lineages, we refrained from analysing the diversity dynamics of Paleorrhyncha or Prosorrhyncha. Furthermore, we considered that even though the monophyly of 'Fulgoridiidae' is doubtful, their placement within Auchenorrhyncha is clearly established and supported by synapomorphies shared by this family and the other Auchenorrhyncha[105]. The monophyly issue encountered with some extinct groups of Hemiptera is probably related to their status of stem lineages blurring their limits with respect to other closely related clades. Hopefully, new hemipteran fossils (especially, early occurrences from Carboniferous and Permian deposits) will improve the systematic delimitations of these lineages.

## Conclusions and perspectives

Our analyses provide the first comprehensive study of Hemiptera past diversity dynamics, leveraging both the fossil record and recent progress in modelling origination and extinction rates while addressing inherent biases in the fossil record. Our study reveals that the P–T mass extinction event led to the first major turnover in Hemiptera evolutionary history and was followed by a rapid diversification of the main Hemiptera lineages during the Triassic and Jurassic periods, likely driven by the colonisation of vacant ecological niches. A second significant Hemiptera turnover occurred during the Cretaceous, coinciding with the rise and diversification of Angiosperms. This period saw the extinctions of relict Paleozoic and many Mesozoic taxa and the diversification of modern hemipteran lineages[32,63].

Our study also identifies potential correlations between Hemiptera diversification and floral changes in past environments. These shifts may have either fuelled Hemiptera diversification by opening new ecological niches, buffering them against extinction, or favoured their extinction during rapid and major floral turnover such as the ATR. Additionally, we highlight the role of diversity dependence—interactions such as predation and competition within and between hemipteran lineages—that may have constrained their diversification throughout evolutionary history.

We propose that the early diversification of Heteroptera during the Triassic could have been driven by major changes in freshwater ecosystems after the P–T crisis. The mass extinction likely vacated many ecological niches, facilitating the rise of new aquatic predators (Nepomorpha) and prey

species with aquatic larval stages[106–109] (such as Coleoptera, Diptera or Trichoptera). Although promising, this hypothesis remains to be tested in future studies.

It has been hypothesised that the Oligocene–Miocene global cooling and drying led to the radiation and expansion of grasslands[110–112], potentially influencing the evolutionary history of cicadomorphan lineages (mainly the leafhoppers and treehoppers), which colonised these habitats[56,113]. Our results suggest that this event only slightly stimulated Hemiptera diversification, particularly within Auchenorrhyncha during the Miocene. However, the evolutionary history of Auchenorrhyncha is intricate, with numerous extinct lineages and recent radiations.

Therefore, we advocate for the development of a molecular counterpart to this study, specifically through the use of phylogenetic birth-death models (e.g. Bayesian Analysis of Mixture Models[114]; RPanda[115]). These models, which are frequently employed to explore the diversity dynamics of insect lineages[116,117], remain underutilised to study Hemiptera evolutionary history. Nevertheless, they are particularly valuable as they can identify shifts in diversification rates at different points in time and across the phylogeny, associating these shifts with specific clades (referred to as macroevolutionary cohorts with shared diversification regimes). To embrace this vision, the acquisition of new genomic data is essential to resolve deep nodes in the Hemiptera tree of life[36]. Additionally, a more comprehensive sampling, beyond what was available in previous studies, will be necessary. We believe that this approach could complement the conclusions derived from the fossil record.

Additionally, tree-based approaches are essential for studying lineages with sparse or nonexistent fossil records, such as ghost lineages (e.g. Enicocephalomorpha). These methods can also provide insights into the complex evolutionary history of Auchenorrhyncha, which requires integrating well-sampled phylogenies with fossil data. Such an integrated approach is vital for unravelling their evolutionary trajectory and assessing the impact of key events on their diversification. Ultimately, combining neontological and paleontological data in joint analyses will enhance their reciprocal illumination, clarifying taxonomic relationships, the understanding of lineage diversity dynamics, and the evaluation of the role of key innovations[118,119].

Our study offers a holistic perspective on Hemiptera diversification, drawing on the fossil record and linking their evolutionary trajectory to crucial biotic factors, such as shifts in global floral assemblages and diversity dependence. It sets the foundation for future analyses of Hemiptera evolution in deep time, shedding light on some of the intricate forces that have influenced their long-term success.

## Methods
### Fossil record of Hemiptera
**Fossil occurrences.** We initially extracted the fossils occurrences of Hemiptera from The Palaeobiology Database (PBDB, https://paleobiodb.org/; downloaded in March 2022). All synonyms, outdated combinations, *nomina dubia* and other erroneous and doubtful records, were corrected. Numerous occurrences of fossils attributed to species were revised to genus- or family-level occurrences. We surveyed the bibliography and visited institutional collections to enhance our dataset. The final dataset encompasses 11,842 fossil occurrences for 244 families and 1794 genera ranging from Pennsylvanian (Carboniferous) to Holocene (compilation stopped on 28th February 2024). Compared to the data available for Hemiptera in PBDB (accessed on 23rd March 2024), we doubled the number of genus-level, family-level and total occurrences.

**Systematic framework.** We followed the most comprehensive systematic framework for Hemiptera to date, integrating extant and extinct lineages[32]. Additionally, we consider that Fulgoromorpha and Cicadomorpha *sensu* Szwedo[32] can be treated as Auchenorrhyncha, a long-lasting monophyletic unit[36,120,121] to aggregate data and reconstruct common palaeodiversity patterns. For Carboniferous hemipterans, we disagreed with Bucher et al.[122] placing *Aviorrhyncha magnifica* Nel,

Bourgoin, Engel and Szwedo 2013 as a Fulgoromorpha, *contra* Szwedo[32]; the specimen was assigned to Hemiptera based on the cubitus anterior vein ending into the radius+media stem far from wing base (plesiomorphy), the lack of a good imprint of the basal and anal parts prevents any better classification. Furthermore, the re-examination of the holotype of the earliest representative of Hemiptera, *Protoprosbole straeleni* Laurentiaux, 1952, allowed us to assign it to Auchenorrhyncha based on the presence of an ambient vein and a postcubital vein with two branches. In addition to published data, we added unpublished occurrences identifiable by 'gen. nov.' or 'fam. nov.' on the datasets. These occurrences have often been known for decades such as the first representative of the subfamily Holoptilinae (Heteroptera: Reduviidae) found in Eocene Oise amber[123].

**Ages.** Occurrences herein are specimens originating from a given stratigraphic horizon assigned to a given taxon. The age of each occurrence is based on data from PBDB. Still, we standardised the occurrences at stage-level precision, when possible, using the stratigraphic framework proposed in the International Chronostratigraphic Chart latest version[124] (ICS 2023/09).

Hemipteran fossils are found in deposits of contentious ages such as the Miocene Dominican amber or the Eocene Baltic amber. For instance, the Miocene Dominican amber is mined in different deposits ranging from Langhian to Burdigalian ages, thus we kept this time range as we often lack accurate mining information for these fossils[125,126]. The same problem occurs for the occurrences within pieces from mid-Cretaceous Kachin and Baltic ambers. The material provenance is not always documented. The mid-Cretaceous Kachin amber, is mined mostly from Noije Bum mines, and was accurately dated to the mid-Cretaceous[127] (U–Pb dating). However, there is often no data about the mines from which the amber pieces originated, rendering it difficult to accurately date the fossil material. To account for these uncertainties, we assigned a Cenomanian age to all fossil occurrences from Kachin amber.

**Systematic datasets.** Hemipteran major lineages originated and diversified at least during the Late Carboniferous[34,36]. We defined an initial global Hemiptera dataset including all the fossils attributed to this order. However, we suspected that the dynamics within and between the major lineages of Hemiptera could be significantly heterogeneous[23]. Consequently, we subsampled our total dataset to explore this putative heterogeneity and constructed three sub-datasets: 'Auchenorrhyncha' (cicadas and hoppers); 'Heteroptera' (true bugs) and 'Sternorrhyncha' (aphids). The fossil record of the sub-order Coleorrhyncha, with numerous putative late Paleozoic occurrences, was judged too lacunar to be confidently analysed separately ($n = 141$ occurrences; without occurrences through the Cenozoic era, encompassing 14 genera and three families); the group is considered to be a relict since the Permian (see Supplementary Figs. 8 and 13).

We focused on genus-level and family-level analyses to depict the diversity dynamics. This approach could be criticised[128]. However, several occurrences of Paleozoic Hemiptera lineages are poorly defined at the species level, complicating analysis at this level. Additionally, the genus level is more resilient to stratigraphic binning and often more taxonomically stable than the species level[129,130]. Fossil insect species are nearly always described from one deposit or one specific geological stage. This clustering into geological stages is particularly problematic as the diversity present in a given stage is wiped out when transitioning to the next stage resulting in an artificial extinction event. This pattern can be partially explained by the short lifespan of insect species (ca. 5 Ma, with some cases of longer lifespans up to 45 Ma; see Grimaldi and Engel[49], p. 15, Table 14.2), related to a short development time and numerous offspring, favouring rapid speciation process within insects. When conducting analyses at the genus level or family level this bias is mitigated and artificial mass extinction events are eliminated. Analyses performed at higher taxonomic levels, such as genus or family, are also particularly valuable for diminishing the prevalence of

singletons. This is because genera or families are typically distributed across multiple geological stages, resulting in a greater number of occurrences[23]. Lastly, genus- and family-level analyses facilitate the inclusion of occurrences that cannot be assigned to a specific species but can be confidently classified at a higher taxonomic rank. This approach enables more accurate modelling and estimation of lineage extinction. For example, the last record of a family may not correspond to a formally named species but could be a partial specimen, potentially unnamed due to preservation issues. A species-level analysis would overlook such occurrences.

All genus-level datasets were extracted from our global dataset and are available in Supplementary Files 1 and 2, and deposited in the Figshare digital data repository (https://figshare.com/s/89f243cf9bab1964e766).

## Dynamics of origination and extinction through time

We analysed our datasets using the Bayesian framework implemented in the programme PyRate 3.8[27,28] to estimate the times of speciation ($T_s$) and extinction ($T_e$) for each taxon and the temporal dynamics of origination ($\lambda$) and extinction ($\mu$) rates.

All analyses were set under the best-fitting preservation process after comparing the following models[27] (-PPmodeltest option): the homogeneous Poisson process (-mHPP option), the non-homogeneous Poisson process (-default option) and the time-variable Poisson process (-qShift option). The time-variable Poisson process assumes that the preservation rates (expected number of occurrences *per lineage per* Ma) are constant within a predefined time frame but may vary over time. In the present study, we used a predefined time bins corresponding to geological stages since the Bashkirian (Carboniferous stage; 323.2–315.2 Ma). This model is thus appropriate when rates are heterogeneous over time. In addition to the time-dependent heterogeneity of preservation rates, we modelled lineage-dependent heterogeneity of preservation rates using a Gamma model[27] (-mG option), this model assumes that preservation rates vary according to a Gamma distribution approximated by four discrete rate categories. As the fossil record of Hemiptera is spatially and temporally heterogeneous, possibly affected by a Lagerstätte effect, and hemipteran lineages display various ecological behaviours leading to putative taphonomical biases, the time-variable Poisson process + Gamma model offers the best approach to analyse our datasets (see Supplementary File 4).

We estimated shifts in origination and extinction within Hemiptera under two different models for each dataset: the BDCS model[76] and the RJMCMC model[27] (-A 4 option). This dual approach allows us to estimate the past diversity dynamics of Hemiptera through time and the parameters of interest over time bins defined a priori (BDCS) or in a more relaxed framework (RJMCMC). The BDCS analyses were conducted with (I) 20-Ma-bins (to smooth the heterogeneity of the fossil record of Hemiptera) and (II) 5-Ma-bins as a sensitivity analysis to detect fast changes (especially around extinction events, see Fig. 1). The RJMCMC analyses were set with time bins corresponding to geological stages.

Estimating the diversification of Hemiptera using two different models (BDCS and RJMCMC) with different assumptions on prior choices, follows the 'best practices' proposed to investigate diversity dynamics with PyRate[131].

For each analysis, we ran PyRate for 200,000,000 Markov chain Monte Carlo (MCMC) iterations with a sampling frequency of 1000 and combined the posterior samples of the parameters from the 10 randomly replicated datasets after excluding the first 10% of the samples as burnin. These 10 datasets were generated to randomly resample the age of fossil occurrences within their respective temporal ranges (*extract.ages* function). We monitored *a posteriori* chain mixing and examined the effective sample sizes (ESS) with Tracer 1.7.1[132]. We consider that the parameters are convergent when ESS are greater than 200. The posterior estimates of the origination and extinction rates across all replicates were combined to generate rate-through-time plots for the origination rates ($\lambda$), the extinction rates ($\mu$) and the net diversification rates (defined as the origination minus extinction). We also estimated the number of lineages through time using the ten posterior estimates of the $T_s$ and $T_e$ for all genera and families (-ltt 1 option).

A period of diversification is characterised by a net diversification rate superior to 0 ($\lambda > \mu$) while a period of extinction is marked by a net diversification rate inferior to 0 ($\lambda < \mu$). We considered shifts of diversification significant when the log Bayes factors are superior to 6 with the RJMCMC model, while we considered shifts significant, under the BDCS model, when mean rates in a time bin do not overlap with the 95% CI of the rates of adjacent time bins.

## Correlations with abiotic and biotic variables

We used the MBD model[133] to assess to what extent biotic and abiotic factors can explain temporal variation in the origination and extinction rates of Hemiptera. We performed our analyses using the $T_s$ and $T_e$ estimated in our previous RJMCMC analyses. The MBD model allows for origination and extinction rates to change through time, related to changes in environmental variables so that origination and extinction rates depend on the temporal variations of each factor. The strength and sign (positive or negative) of the correlations are jointly estimated for each variable. A horseshoe prior was used to reduce the risk of over-parameterisation[133]. The seven variables: 'Diversity-dependence', 'Angiosperm', 'Gymnosperm', 'Polypodiales', 'Spores-Plants', 'Temperature' and 'Non-Polypodiales', were selected because of their potential to be linked with insect diversity dynamics[22-25].

The fluctuation in the relative diversity of different plant lineages has likely driven the diversification of numerous insect lineages[134-136]. Interactions between insects and plants have been documented since the Carboniferous[137-139], some cylindrical holes in Carboniferous plants have been associated with damages produced by piercing-and-sucking insects, potentially Palaeodictyopterida, Thripida and Hemiptera[140].

The Hemiptera are a large clade of piercing-sucking insects that have likely evolved with floral changes and therefore have undergone shifts in their host plants[37]. These host plant switches may have exerted strong selection pressures on phytophagous hemipteran lineages. The data of different plant lineages (with 1-million-year time intervals, and scaled to vary between 0 and 1) are from Silvestro et al.[76] and Lehtonen et al.[133]: Angiosperms, Gymnosperms, Spore plants (ferns, liverworts, mosses, lycophytes) and extracted from the latter, the relative diversity of non-Polypodiales ferns and Polypodiales ferns. These diversity trajectories were modelled based on comprehensive fossil record compilations, using a birth-death model with rate shifts defined by the epochs of the stratigraphic geological timescale, and time-calibrated phylogenies based on molecular data[76,133]. These diversity trajectories were modelled while accounting for uncertainties in the fossil record, and variations in preservation rates over time, and are considered to be relatively robust. Discovery of new fossils might require an update of these estimates, but global variations are expected to be stable.

Deep-time climatic variations (warming/cooling periods) likely correspond to a driver of the fluctuations in insect diversity[20,141]. Temperature variations through time may have acted as a factor, influencing both the diversity of Hemiptera and their food resources, especially plants[89]. Global climate change patterns over time are commonly estimated using the relative proportions of different oxygen isotopes ($\delta^{18}O$) found in samples of benthic foraminifera shells[142]. To assess temperature variations, we utilised a dataset of absolute temperatures compiled by Condamine et al.[143] for the last 500 million years, at 1-million-year intervals (see their section on *Global temperature variations through time*). These temperature data reflect planetary-scale climatic trends—which are relatively well-established compared to local fluctuations (in deep time) and less prone to major revisions—and are based on $\Delta^{18}O$ data measured from benthic foraminifer shells preserved in oceanic sediments from Cramer et al.[144], Prokoph et al.[145] and Zachos et al.[142,146]. Additionally, diversity-dependency effects on the diversification or extinctions within Hemiptera are suspected, notably through competition[94]. We investigated the hypothesis of diversity dependence between Hemiptera lineages using the MBD model, and the diversity through time (i.e. lineage-through-time value) of each clade (i.e. Hemiptera, Auchenorrhyncha, Sternorrhyncha and Heteroptera) estimated with the RJMCMC model—to assess to what extent the variation in diversity of these

lineages correlates with the origination and extinction rates of the other. We followed a hypothetico-deductive approach and we selected the variables to verify biological hypotheses proposed in the literature or raised by our study[147]. All the datasets are available in Supplementary File 5.

We ran our MBD analyses for 50 million MCMC iterations, with a sampling frequency of 50,000, for all datasets at genus and family level (see Supplementary Tables 1–8 for the latter). The visualisation of results and convergence parameters was performed with Tracer 1.7.1[132]. We considered correlation parameters to be significant when the shrinkage weights ($\omega$) are strictly superior to 0.5 and the 95% CI does not overlap with 0.

## Reporting summary

Further information on research design is available in the Nature Portfolio Reporting Summary linked to this article.

## Data availability

All data generated in this study have been deposited in a Figshare digital data repository (https://doi.org/10.6084/m9.figshare.c.7530903) (ref. 148).

## Code availability

PyRate and PyRateMBD codes are available in a Zenodo archive (ref. 149). The specific command lines set to run all the models presented in this study are available in a Figshare digital data repository (https://figshare.com/s/89f243cf9bab1964e766) (ref. 148).

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

## Acknowledgements
We thank Dr. Frédéric Legendre (MNHN) who generously provided insightful comments and suggestions on the first version of our manuscript; Dr. Iwan Stössel (ETH) and Dr. Diying Huang (NIGPAS) who kindly gave us access respectively to ETH Zürich and NIGPAS collections, helping to precise the taxonomic accuracy of reported and unreported occurrences; and Dr. Joseph T. Flannery-Sutherland (University of Birmingham) for providing the R script adapted for the MBD results visualisation. We are grateful to the PBDB team and contributors for their constant effort in maintaining their database and for making it freely available. This work is part of MB's Ph.D. thesis.

## Author contributions
Mathieu Boderau (conceptualisation, investigation, data curation, formal analysis, writing—original draft, writing—review and editing), André Nel (supervision, conceptualisation, investigation, data curation, writing—original draft, writing—review and editing) and Corentin Jouault (supervision, conceptualisation, investigation, data curation, formal analysis, writing—original draft, writing—review and editing).

## Competing interests
The authors declare no competing interests.
