## [Transparent Peer Review file · Communications Biology]

Diversification and extinction of Hemiptera in deep time

Corresponding Author: Mr Mathieu Boderau

This manuscript has been previously reviewed at another journal. This document only contains information relating to versions considered at Communications Biology.

Version 0:

Reviewer comments:

Reviewer #1

(Remarks to the Author)

This is a lovely paper and was a pleasure to review. Sorry for the delay, it was winter break. I assumed the deadline might be extended but received an email on boxing day. I only came back to work on Monday and have finished the review in the past few days.

Very well done. I like the statistical approach and your conclusions. Furthermore, your writing style is very nice. A nice mix of "prose" as well as scientific writing makes it an easy read. I recommend this for publication following refining and revisiting a few things.

I have a few suggestions, but very few of them require major changes:

- Not sure "hypothesis-driven approach" is needed in the abstract
- Phylogenies are not necessarily bad, and Louca and Pennell were talking about a specific subset of models
- Please check debates following Louca and Pennell, in order to comment on whether people have investigated your question with phylogenies before, and whether they agree with your results. <https://doi.org/10.1093/sysbio/syab083>, [https://www.cell.com/trends/ecology-evolution/abstract/S0169-5347\(22\)00027-1](https://www.cell.com/trends/ecology-evolution/abstract/S0169-5347(22)00027-1), <https://doi.org/10.1111/2041-210X.14240>, <https://doi.org/10.1111/2041-210X.13997> (the latter used here <https://doi.org/10.1098/rsbl.2023.0314>)
- There are 107,000 extant species but only 3,350 extinct. Can you talk more about the phylogenetic distribution of these extinct species? Do they fit into extant lineages? To the general reader with interest (but perhaps not expertise), this information is crucial
- Here is also a good time to talk about what phylogenies can actually tell us. I do appreciate their inferences should definitely be scrutinised and questioned. But they also provide a framework to study the massive swathes of biodiversity that did not fossilise. The fossil record is definitely not infallible either (<https://besjournals.onlinelibrary.wiley.com/doi/full/10.1111/2041-210X.12666>) and fossils and phylogenies should be interrogated in conjunction in a holistic way, please check debates in <https://doi.org/10.1111/nph.15708>, <https://doi.org/10.1098/rsbl.2023.0314>, <https://doi.org/10.1098/rsbl.2024.0265>, <https://doi.org/10.1002/ajb2.16282>. This paper in particular is very good <https://doi.org/10.1111/nph.19010>
- I do appreciate that the scale of hemiptera phylogenetics makes this currently impossible. But it could be very good to discuss this and pre-empt it in future research, and guide understanding of the nuances of this debate. It could elevate the impact of your paper greatly
- The information in lines 74-93 are interesting, but their relevance arrives too late. Could you flip the sentences around, so the impact on evolution (the purpose) arrives sooner?
- Line 101-105 seem too relevant and interesting to be relegated to the end of the section
- Lines 106-110 are also interesting but seem a bit redundant, or at least poorly integrated
- Recent work has looked at impacts of global cooling on diversification dynamics, such as <https://www.nature.com/articles/ncomms13003>, <https://www.pnas.org/doi/abs/10.1073/pnas.2102408120>, <https://www.biorxiv.org/content/10.1101/392712v1.abstract>
- In line 124, are there 11,840 species or occurrences?
- Lines 151-152 are confusing, why mention the 20 ma time bins analysis? It seems like the higher resolution analysis is better
- What does studying all three of species, genus and family levels tell us that studying just species level does not? I know the reason, but readers might not. You could refer to studies of angiosperms, since their fossil record is poor, people often look at higher lineage dynamics

- Could you explain within the text how the variables of angiosperm (and other plants) are calculated?
- Your figures are really nice and clear, well done!
- Is it possible to plot a phylogeny (e.g. at family level or something), and show how many extant species there are compared to extinct species in your sample at tips? It would be great to visualise this, and very informative on a lower level (just getting the reader's head around the sample) and higher level (understanding taxonomic bias)
- Lovely discussion section, very interesting and well-referenced
- In lines 283-285, please refine this and read debates on the floral evolution across K-Pg in plants in <https://doi.org/10.1111/nph.13247>, <https://doi.org/10.1098/rsbl.2023.0314>, <https://doi.org/10.1111/ter.12086>, <https://www.science.org/doi/full/10.1126/science.abf1969>, <https://doi.org/10.1017/ext.2023.13>, <https://doi.org/10.1016/j.revpalbo.2023.104933>. The debate on the impact of K-Pg on plants is much more nuanced, e.g. different processes at different levels, and would provide some very interesting things to discuss! I would love to hear your thoughts on this
- Lines 448-452 could also benefit from reading these papers. I am sure there are other examples too, but there is debate on the impacts on environmental changes (K-Pg) at different taxonomic levels in these papers
- Good to cite Peris and Condamine, 2024
- Line 493, "As for the a gymnosperm" does not make sense
- Good to include an explicit limitations paragraph, I appreciate the deep discussion on this
- The methods section is great. But could you explain how the different variables in the MBD were defined and calculated? Are there any limitations to this data?

Reviewer #2

(Remarks to the Author)

With pleasure, I read this manuscript describing a complex issue like as diversification and extinction of Hemiptera. It's an excellent paper, unique and valuable. This paper is properly organised, the title is informative and concise, and the abstract is representative of the content. The authors prepare an introduction based on relevant references. Authors based on the huge material and modern methods during their analysis. Methods are well described, and all-sufficient methodological detail was that the experiments could be reproduced; all figures are necessary and well-labelled. The discussion and conclusion are supported by the results, and the results are clearly presented. All claims are convincing, and appropriately discussed in the context of previous literature. It is a huge article collecting information about the entire order of insects, discussing all groups of insects included in it and I'm sure that this article will clarify some previously controversial issues. It is a very important and helpful paper for the next researchers who will analyze a group of hemipteran species in the future. I think, that the paper will be of interest to others in the field, not only for hemipterologists and maybe not only entomologists at all.

The manuscript is clearly written. I am not qualified to assess the quality of English in this paper, but the language is precise and understandable also for non-native speakers.

I have a few questions or suggestions and a lot of corrections in References.

Line 74. Hemiptera are characterized by a segmented rostrum with a multi-segmented sheet-like labium covering the mandibular and maxillary stylets (Emeljanov, 2002).

This sentence is not clear: only the labium is segmented; the rest parts of the mouthparts aren't segmented in Hemiptera. I suggest citing other papers, more modern (for example Brožek J. et al. 2019, 2020, 2022), not only Emeljanov, 2002.

Line 81. What taxonomic system use authors in this paper? Which one? After whom?

Lines 87-93. Heteroptera switched to phytophagy several times, but each time differently; it was not just a return to phloem (in some, yes) but to fruits, seeds, algae, fungi, etc. (Panizzi et al. 2021).

Line 106. The oldest Nepomorpha are Triassic, so they did not re-enter the water, but settled freshwater during the Jurassic Lacustrine revolution, similarly Gerromorpha probably then became widespread.

Line 118. During the Oligocene and Miocene, there were mainly open areas - herbs, then there was a change in metabolism from C4 to C3 in grasses hence the shift and transition in some to other endosymbionts, hence the radiation of Delphacidae and Cicadellidae. Also in some of the aphids that like colder climates and also some of the maggots associated with grasses.

References corrections:

Papers not cited in the text: Alroy, J. (2016); Barnosky, A.D. (1999).; Barnosky, A.D. (1999); Bernardi, M., Gianolla, P., Petti, F.M., Mietto, P. and Benton, M.J. (2018); Condamine, F.L., Rolland, J. and Morlon, H. (2019); Depa, L., Kaszyca-Taszakowska, N., Taszakowski, A. and Kanturski, M. 2020; Foote, M. and Miller I.A. (2007); Lovegrove, B.G. and Mowoe, M.O. (2015); Nel, A. (1997); Popov, Y.A. (1968); Van Valen, L. 1973. Lack of references: Louca and Pennell, 2020; Li et al., 2017; Labandeira and Li, 2021; Burckhardt et al., 2022; Labandeira, 1997; Prokop et al., 2019; Rainford et al., 2014.

Different years in the text and references: Pyron and Burbrink, 2013; Jiang et al., 2022; Jouault et al., 2024 a or b? Wrong order, not alphabetical, for example, Cryan/ Condamine.

In my opinion, the overall quality of the work is high and I warmly recommend this article for publication in Communications Biology after small corrections.

Version 1:

Reviewer comments:

Reviewer #1

(Remarks to the Author)

Thank you for making revisions. I am satisfied with this paper as-is and recommend it for publication. It was already a very solid paper, but the revisions I think have made it stronger. I especially like the phylogeny figure showing numbers of genera sampled compared to the extant richness. This looks great. The clarifications I asked for have been satisfactorily addressed too. My only concern is the title, which ends "in the deep time". Is the "the" needed here? I've only really read "deep time" before, rather than "the deep time".

Congratulations, a very cool paper. I hope to see it in print soon!

Reviewer #2

(Remarks to the Author)

Thank you to the authors for taking all my suggestions into account. I have no additional comments. In my opinion, the article is suitable for publication in *Communication Biology* in its present form.

Reviewer #1 (Remarks to the Author):

This is a lovely paper and was a pleasure to review. Sorry for the delay, it was winter break. I assumed the deadline might be extended but received an email on boxing day. I only came back to work on Monday and have finished the review in the past few days.

Very well done. I like the statistical approach and your conclusions. Furthermore, your writing style is very nice. A nice mix of “prose” as well as scientific writing makes it an easy read. I recommend this for publication following refining and revisiting a few things.

Thank you for reviewing our manuscript and the positive feedback.

I have a few suggestions, but very few of them require major changes:

- Not sure “hypothesis-driven approach” is needed in the abstract

Thank you for your comment. We have revised the sentence to highlight the Bayesian methodology (approach) and the Hemiptera fossil record (data) more prominently.

- Phylogenies are not necessarily bad, and Louca and Pennell were talking about a specific subset of models
- Please check debates following Louca and Pennell, in order to comment on whether people have investigated your question with phylogenies before, and whether they agree with your results.
<https://doi.org/10.1093/sysbio/syab083>, [https://www.cell.com/trends/ecology-evolution/abstract/S0169-5347\(22\)00027-1](https://www.cell.com/trends/ecology-evolution/abstract/S0169-5347(22)00027-1), <https://doi.org/10.1111/2041-210X.14240>,
<https://doi.org/10.1111/2041-210X.13997> (the latter used here
<https://doi.org/10.1098/rsbl.2023.0314>)

Thank you for pointing out the blunt nature of our phrasing. Our intention was not to overemphasize the work of Louca and Pennell (2020), as we recognize the ongoing efforts to address the potential for ‘identifiability issues.’ We have rephrased this section to highlight the progress being made. We strongly believe that adopting hypothesis-driven approaches, incorporating priors within Bayesian frameworks, and penalizing for model complexity can help mitigate identifiability issues and enhance the effective application of existing diversification methods (as highlighted in Rabosky et al., 2017; Rabosky, 2018; Morlon et al., 2022).

References:

Rabosky, D.L., Mitchell, J.S. and Chang, J. (2017) Is BAMM Flawed? Theoretical and Practical Concerns in the Analysis of Multi-Rate Diversification Models. *Systematic Biology*, 66(4), 477–498. <https://doi.org/10.1093/sysbio/syx037>

Rabosky, D.L. (2018) BAMM at the court of false equivalency: A response to Meyer and Wiens. *Evolution*, 72(10), 2246– 2256. <https://doi.org/10.1111/evo.13566>

Louca, S. and Pennell, M.W. (2020) Extant timetrees are consistent with a myriad of diversification histories. *Nature*, 580: 502–505. <https://doi.org/10.1038/s41586-020-2176-1>

Morlon, H., Robin, S. and Hartig, F. (2022) Studying speciation and extinction dynamics from phylogenies: addressing identifiability issues. *Trends in Ecology and Evolution*, 37(6), 497– 506. <https://doi.org/10.1016/j.tree.2022.02.004>

- There are 107,000 extant species but only 3,350 extinct. Can you talk more about the phylogenetic distribution of these extinct species? Do they fit into extant lineages? To the general reader with interest (but perhaps not expertise), this information is crucial.

We agree that this information is valuable for a general audience. At the species level, the majority of Hemiptera fossils belong to the extant major lineages: Heteroptera (true bugs), Auchenorrhyncha (true hoppers), and Sternorrhyncha (aphids or scale insects). Only 145 fossil species remain unassigned to these lineages. To offer readers a clearer understanding of the sampling distribution across both extinct and extant major Hemiptera lineages, we have incorporated detailed information about this distribution within the tree (that you ‘requested’ in your subsequent comment).

- Here is also a good time to talk about what phylogenies can actually tell us. I do appreciate their inferences should definitely be scrutinised and questioned. But they also provide a framework to study the massive swathes of biodiversity that did not fossilise. The fossil record is definitely not infallible either (<https://besjournals.onlinelibrary.wiley.com/doi/full/10.1111/2041-210X.12666>) and fossils and phylogenies should be interrogated in conjunction in a holistic way, please check debates in <https://doi.org/10.1111/nph.15708>, <https://doi.org/10.1098/rsbl.2023.0314>, <https://doi.org/10.1098/rsbl.2024.0265>, <https://doi.org/10.1002/ajb2.16282>. This paper in particular is very good <https://doi.org/10.1111/nph.19010> I do appreciate that the scale of hemiptera phylogenetics makes this currently impossible. But it could be very good to discuss this and pre-empt it in future research, and guide understanding of the nuances of this debate. It could elevate the impact of your paper greatly

Thank you for your comment; we completely agree with your perspective. However, we believe that the article format does not allow sufficient space to thoroughly discuss the integration of paleontological and neontological data and their reciprocal illumination, particularly within the *Introduction*, as it diverges from the main focus of our paper. That said, we have expanded the *Conclusions and Perspectives* sections to highlight our view on how the Hemiptera fossil record can be enriched by combining it with molecular data. We also provide examples to illustrate how such integration could enhance our understanding of Hemiptera diversity dynamics and, more broadly, macroevolutionary processes.

We added the following paragraphs: « *Therefore, we advocate for the development of a molecular counterpart to this study, specifically through the use of phylogenetic birth-death models (Bayesian Analysis of Mixture Models: Rabosky et al., 2014; RPanda: Morlon et al., 2016). These models, which are frequently employed to explore the diversity dynamics of insect lineages (Blaimer et al., 2023; Kawahara et al., 2023), remain underutilized to study Hemiptera evolutionary history. Nevertheless, they are particularly valuable as they can identify shifts in diversification rates at different points in time and across the phylogeny, associating these shifts with specific clades (referred to as macroevolutionary cohorts with shared diversification regimes). To embrace this vision, the acquisition of new genomic data is essential to resolve deep nodes in the Hemiptera tree of life (Johnson et al., 2018). Additionally, a more comprehensive sampling, beyond what was available in previous studies, will be necessary. We believe that this approach could complement the conclusions derived from the fossil record.*

Additionally, tree-based approaches are essential for studying lineages with sparse or nonexistent fossil records, such as ghost lineages (e.g., Enicocephalomorpha). These methods can also provide insights into the complex evolutionary history of Auchenorrhyncha, which requires integrating well-sampled phylogenies with fossil data. Such an integrated approach is vital for unraveling their evolutionary trajectory and assessing the impact of key events on their diversification. Ultimately, combining neontological and paleontological data in joint analyses will enhance their reciprocal illumination, clarifying taxonomic relationships, the understanding of lineage diversity dynamics, and the evaluation of the role of key innovations (Vea and Grimaldi, 2016; Spasojevic et al., 2021).

Our study offers a holistic perspective on Hemiptera diversification, drawing on the fossil record and linking their evolutionary trajectory to crucial biotic factors, such as shifts in global floral assemblages and diversity dependence. It sets a crucial foundation for future analyses of Hemiptera evolution through deep time, shedding light on some of the intricate forces that have influenced their long-term success. »

References:

Blaimer, B.B., Santos, B.F., Cruaud, A., Gates, M.W., Kula, R.R., Mikó, I., Rasplus, J.-Y., Smith, D.R., Talamas, E.J., Brady, S.G. and Buffington, M.L., (2023) Key innovations and the diversification of Hymenoptera. *Nature Communications*, 14, 1212. <https://doi.org/10.1038/s41467-023-36868-4>

Johnson, K.P., Dietrich, C.H., Friedrich, F., Beutel, R.G., Wipfler, B., Peters, R.S., Allen, J.M., Petersen, M., Donath, A., Walden, K.K.O., Kozlov, A.M., Podsiadlowski, L., Mayer, C., Meusemann, K., Vasilikopoulos, A., Waterhouse, R.M., Cameron, S.L., Weirauch, C., Swanson, D.R., Percy, D.M., Hardy, N.B., Terry, I., Liu, S., Zhou, X., Misof, B., Robertson, H.M. and Yoshizawa, K. (2018) Phylogenomics and the evolution of hemipteroid insects. *Proceedings of the National Academy of Sciences of the United States of America*, 115 (50), 12775–12780. <https://doi.org/10.1073/pnas.1815820115>

Kawahara, A.Y., Storer, C., Carvalho, A.P.S. *et al.* (2023) A global phylogeny of butterflies reveals their evolutionary history, ancestral hosts and biogeographic origins. *Nature Ecology & Evolution*, 7, 903–913. <https://doi.org/10.1038/s41559-023-02041-9>

Morlon, H., Lewitus, E., Condamine, F.L., Manceau, M., Clavel, J. and Drury, J. (2016) RPANDA: an R package for macroevolutionary analyses on phylogenetic trees. *Methods in Ecology and Evolution*, 7, 589–597. <https://doi.org/10.1111/2041-210X.12526>

Rabosky, D.L. (2014) Automatic detection of key innovations, rate shifts, and diversity-dependence on phylogenetic trees. *PLoS ONE* 9, e89543. <https://doi.org/10.1371/journal.pone.0089543>

Spasojevic, T., Broad, G.R., Sääksjärvi, I.E., Schwarz, M., Ito, M., Korenko, S. and Klopstein, S. (2021) Mind the Outgroup and Bare Branches in Total-Evidence Dating: a Case Study of Pimpliform Darwin Wasps (Hymenoptera, Ichneumonidae). *Systematic Biology*, 70, 322–339. <https://doi.org/10.1093/sysbio/syaa079>

Vea, I. and Grimaldi, D. (2016) Putting scales into evolutionary time: the divergence of major scale insect lineages (Hemiptera) predates the radiation of modern angiosperm hosts. *Scientific Reports*, 6, 23487. <https://doi.org/10.1038/srep23487>

- The information in lines 74-93 are interesting, but their relevance arrives too late. Could you flip the sentences around, so the impact on evolution (the purpose) arrives sooner?

Thank you for your feedback. We have restructured this paragraph, along with the subsequent ones, to place greater emphasis on the “impact on evolution.” This reorganization also ensures better alignment with the opening of the following paragraph.

- Line 101-105 seem too relevant and interesting to be relegated to the end of the section

Thank you for your comment. We have reorganized this paragraph to focus more directly on the Mid Mesozoic Parasitoid Revolution (MMPR) and the role Hemiptera played in this event. In particular, we now begin the paragraph on this concept.

- Lines 106-110 are also interesting but seem a bit redundant, or at least poorly integrated

We thank the reviewer for their insightful comment. During the writing of the article, we debated whether to retain this paragraph. We believe the reviewer’s suggestion aligns with the decision to remove it, as it enhances the clarity of the manuscript. The paragraph previously introduced “the return to aquatic life in Heteroptera,” a topic that was not explored in detail elsewhere in the manuscript, which could have confused the reader. As a result, we have decided to delete this short paragraph.

- Recent work has looked at impacts of global cooling on diversification dynamics, such as <https://www.nature.com/articles/ncomms13003>, <https://www.pnas.org/doi/abs/10.1073/pnas.2102408120>, <https://www.biorxiv.org/content/10.1101/392712v1.abstract>

Thank you for suggesting additional references. While we appreciate the suggestion, we feel that Davies *et al.* (2016 - Nature Communications), which focuses on Anomura (a marine clade), is not directly relevant to the diversification of a terrestrial clade, so we chose not to cite it. However, we have incorporated the references to Thompson *et al.* (2023) and Davies *et al.* (2018 – bioRxiv), as we believe they offer valuable context to our discussion. The revised paragraph is outlined here:

« Finally, the global cooling and drying during the Oligocene-Miocene spurred the expansion of grasslands—favoring to the diversification of certain plant lineages (orchids: Thompson *et al.*, 2023)—which likely contributed to the emergence of new specialized Hemiptera faunas composed by many Cicadomorpha families (Dietrich, 1999; Davies *et al.*, 2018; Szwed, 2018) ».

References:

Davies, K.E., Hill, J., Astrop, T.I. and Willis, M.W. (2016) Global cooling as a driver of diversification in a major marine clade. *Nature Communications*, 7, 13003. <https://doi.org/10.1038/ncomms13003>

Davies, K.E., Bakewell, A.T., Hill, J., Song, H., Mayhew, P. (2018) Global cooling & the rise of modern grasslands: Revealing cause & effect of environmental change on insect diversification dynamics. *bioRxiv* [Preprint]. <https://doi.org/10.1101/392712> (Accessed 13 January 2025).

Thompson, J.B., Davies, K.E., Dodd, H.O., Willis, M.A. and Priest, N.K. (2023). Speciation across the Earth driven by global cooling in terrestrial orchids. *Proceedings of the National Academy of Sciences of the United States of America*, 120 (29), e2102408120. <https://doi.org/10.1073/pnas.2102408120>

- In line 124, are there 11,840 species or occurrences?

In the original version of the manuscript, we detailed '11,840 fossil occurrences'. In the *Material and Methods* section, we clarified the definition of "fossil occurrences" as: *Occurrences herein are specimens originating from a given stratigraphic horizon assigned to a given taxon*. Therefore, we are referring to the number of fossil specimens, not the number of species.

- Lines 151-152 are confusing, why mention the 20 ma time bins analysis? It seems like the higher resolution analysis is better

Thank you for your question, and we agree with your observation. Initially, we intended to highlight the significance of the 'short time bins' analysis by comparing the results of both long and short time bin analyses. However, we recognize that this might be confusing for the reader. We have since revised the paragraph to focus on the results of the 'short time bins' analysis, which offers better resolution and a more accurate estimate of diversity dynamics.

- What does studying all three of species, genus and family levels tell us that studying just species level does not? I know the reason, but readers might not. You could refer to studies of angiosperms, since their fossil record is poor, people often look at higher lineage dynamics

Thank you for bringing attention to this important aspect of macroevolutionary methodology. In the previous version of the manuscript, we addressed this point in the *Material and Methods* section (*Systematic Datasets*), where we outlined the rationale behind conducting analyses at the genus or family level. In the new version we added some crucial points to this section (see the text in bold) which will help the readers to better understand our choice:

« We focused on genus-level and family-level analyses to depict the diversity dynamics. This approach could be criticized (Hendricks *et al.*, 2014). However, several occurrences of Paleozoic Hemiptera lineages are poorly defined at the species level, complicating analysis at this level. Additionally, the genus level is more resilient to stratigraphic binning and more taxonomically stable than the species level (Allmon, 1992; Foote, 2000). Fossil insect species are nearly always described from one deposit or one specific geological stage. This clustering of species in geological stages is particularly problematic as the diversity present in a given stage is wiped out when transitioning to the next stage resulting in an artificial extinction event. This pattern can be partially explained by the short lifespan of insect species (ca. 5 Ma, with some cases of longer lifespans up to 45 Ma; see in Grimaldi and Engel, 2005: 15, table 14.2), related to a short development time and numerous offspring, favouring rapid speciation process within insects. When conducting analyses at the genus level or family level this bias is mitigated and artificial extinction events are eliminated. Analyses performed at higher taxonomic levels, such as genus or family, are particularly valuable for

diminishing the prevalence of singletons. This is because genera or families are typically distributed across multiple geological stages, resulting in a greater number of occurrences (Jouault et al., 2022b). Lastly, genus- and family-level analyses facilitate the inclusion of occurrences that cannot be assigned to a specific species but can be confidently classified at a higher taxonomic rank. This approach enables more accurate modeling and estimation of lineage extinction. For example, the last record of a family may not correspond to a formally named species but could be a partial specimen, potentially unnamed due to preservation issues. A species-level analysis would overlook such occurrences. ».

We wish to underscore the conventionality of this practice in paleo-entomology, which commonly involves working at the genus or family level. This approach is adopted, among other reasons, due to the limitations outlined earlier. A relevant example is highlighted by Labandeira & Sepkoski (1993), who advocated for the significance of the family level: “*the rank of family was chosen for several reasons. (i) This taxonomic level has been analyzed in other studies of fossil diversity and seems to correlate well with underlying species diversity. (ii) Families are less susceptible to irregular and biased sampling (...), so that an evolutionary signal is better maintained at this level. (iii) Families of insects, especially extant ones, are reasonably well established through consensus among researchers, whereas fossil species (...) are more idiosyncratically defined and less frequently correspond to good phylogenetic or phenetic units. (iv) Insect families individually possess discrete, often highly stereotyped life habits, and their morphologies directly reflect their trophic guilds, which are informative in diversity studies.*”.

This rationale, supporting analyses carried out at the family level as opposed to the species level, is similarly applicable to analyses conducted at the genus level compared to the species level. Notably, to the best of our knowledge, there are no analyses at the species level based on the fossil record for estimating past speciation and extinction rates within insect lineages.

On a broader note, we believe that working at higher taxonomic levels is crucial for understanding the diversity dynamics (diversification, biogeography,...) of lineages, particularly in insects. Not all fossil specimens can be confidently classified at the species level, but many can be reliably assigned to a family. This distinction is important when studying, for example, the biogeographic history of a clade, as focusing solely on species-level data may overlook valuable information. For instance, consider a clade currently distributed in Australia and the Nearctic, with a fossil occurrence confidently assigned to a genus in the Indomalayan bioregion, but not to a specific species. A species-level analysis would miss the potential migration of this clade between Australia and the Nearctic, passing through the Indomalayan region while informed by the fossil record. In contrast, a genus-level analysis would capture this possibility and allow for its modeling, providing more meaningful insights into the clade’s biogeographic history.

References:

- Allmon, W.D. (1992) Genera in paleontology: definition and significance. *Historical Biology*, 6, 149–158. <https://doi.org/10.1080/10292389209380424>
- Foote, M. (2000) Origination and extinction components of taxonomic diversity: general problems. *Paleobiology*, 26, 74–102. <https://doi.org/10.1017/S0094837300026890>
- Grimaldi D. and Engel M.S. 2005. *Evolution of the insects*. Cambridge and New York: Cambridge University Press. xv + 755 pp.
- Hendricks, J.R., Saupe, E.E., Myers, C.E., Hermsen, E.J. and Allmon, W.D. (2014) The generification of the fossil record. *Paleobiology*, 40(4), 511–528. <https://doi.org/10.1666/13076>
- Jouault, C. Nel, A., Perrichot, V., Legendre, F. and Condamine, F.L. (2022b) Multiple drivers and lineage-specific insect extinctions during the Permo-Triassic. *Nature Communications*, 13, 7512. <https://doi.org/10.1038/s41467-022-35284-4>
- Labandeira, C.C. and Sepkoski, J.J. (1993) Insect diversity in the fossil record. *Science*, 261, 310–315. <https://doi.org/10.1126/science.11536548>

- Could you explain within the text how the variables of angiosperm (and other plants) are calculated? The data for the different plant lineages are from Silvestro et al. (2015) and Lehtonen *et al.* (2017): Angiosperms, Gymnosperms, Spore-plants (ferns, liverworts, mosses, lycophytes), and extracted from the latter, the relative diversity of non-Polypodiales ferns and Polypodiales ferns. These diversity trajectories were modeled based on the fossil record, using a birth-death model with rate shifts defined by the epochs of the stratigraphic geological timescale, and time calibrated phylogenies based on molecular data (Silvestro et al., 2015; Lehtonen et al., 2017). They were rescaled into 1-million-year time intervals, and to vary between 0–1 in the original studies.

We also took advantage of this round of review to provide additional details for other variables, which aligns with your last comment.

References:

- Lehtonen, S., Silvestro D., Karger, D.N., Scotese, C., Tuomisto, H., Kessler, M., Peña, C., Wahlberg, N. and Antonelli, A. (2017) Environmentally driven extinction and opportunistic origination explain fern diversification patterns. *Scientific Reports*, 7, 4831. <https://doi.org/10.1038/s41598-017-05263-7>
- Silvestro, D., Cascales-Miñana, B., Bacon, C.D. and Antonelli, A. (2015) Revisiting the origin and diversification of vascular plants through a comprehensive Bayesian analysis of the fossil record. *New Phytologist*, 207, 425–436. <https://doi.org/10.1111/nph.13247>

- Your figures are really nice and clear, well done!
Thank you for your positive opinion on our figures!

- Is it possible to plot a phylogeny (e.g. at family level or something), and show how many extant species there are compared to extinct species in your sample at tips? It would be great to visualise this, and very informative on a lower level (just getting the reader's head around the sample) and higher level (understanding taxonomic bias)

Thank you for your comment, which aligns with your earlier feedback that we have already partially answered. In response, we have added a simplified Hemiptera Tree of Life to Figure 1 (see panel J), showing the major extant and extinct lineages and their relationships. Next to each clade name, we have indicated the following: the number of genera included in our analyses (in red), the number of occurrences in our datasets (in black), and the number of extant species (in green, based on Streito & Germain, 2020). The figure caption has been updated accordingly to reflect these additions.

Reference:

- Streito, J.-C. and Germain, J.-F. (2020) Chapitre 23: Ordre des Hemiptera (Hémiptères). pp. 481-574. *In: Aberlenc, H.-P. (ed.). Les insectes du monde: biodiversité, classification, clés de détermination des familles.* Museo Editions, Editions Quae: 966 pp.

- Lovely discussion section, very interesting and well-referenced
We greatly appreciate your feedback on our discussion.

- In lines 283-285, please refine this and read debates on the floral evolution across K-Pg in plants in <https://doi.org/10.1111/nph.13247>, <https://doi.org/10.1098/rsbl.2023.0314>, <https://doi.org/10.1111/ter.12086>, <https://www.science.org/doi/full/10.1126/science.abf1969>, <https://doi.org/10.1017/ext.2023.13>, <https://doi.org/10.1016/j.revpalbo.2023.104933>. The debate on the impact of K-Pg on plants is much more nuanced, e.g. different processes at different levels, and would provide some very interesting things to discuss! I would love to hear your thoughts on this

Apologies for the oversight on our part! We did not intend to emphasize the K-Pg event, and the simplest way to address this issue is by removing its mention from the sentence. We fully agree with your perspective that the impact of the K-Pg event on floral assemblages is far from settled, with divergent views and contradictory findings depending on the data and methodologies used to explore this question. This complexity is well illustrated by the recent paper by Thompson and Ramírez-Barahona (2023), the subsequent reply by Hagen (2024), and the counter-reply by Thompson and Ramírez-Barahona (2024).

References:

Thompson, J.B. and Ramírez-Barahona, S. (2023) No phylogenetic evidence for angiosperm mass extinction at the Cretaceous–Palaeogene (K-Pg) boundary. *Biology Letters*, 19, 20230314. <https://doi.org/10.1098/rsbl.2023.0314>

Hagen, E.R. (2024) A critique of Thompson and Ramírez-Barahona (2023) or: how I learned to stop worrying and love the fossil record. *Biology Letters*, 20, 20240039. <https://doi.org/10.1098/rsbl.2024.0039>

Thompson, J.B. and Ramírez-Barahona, S. (2024) The meaning of mass extinctions and what the fossil record tells us about angiosperm survival at K-Pg: a reply to Hagen (2024). *Biology Letters*, 20, 20240265. <http://doi.org/10.1098/rsbl.2024.0265>

- Lines 448-452 could also benefit from reading these papers. I am sure there are other examples too, but there is debate on the impacts on environmental changes (K-Pg) at different taxonomic levels in these papers. Good to cite Peris and Condamine, 2024

We are sorry but it seems that the line there is an inconsistency between our line numbering and the one of the reviewer.

- Line 493, “As for the a gymnosperm” does not make sense

Sorry for the mistake, we rephrased this sentence.

- Good to include an explicit limitations paragraph, I appreciate the deep discussion on this

Thank you for your positive feedback on this paragraph. We believe it is essential to clearly outline the limitations of our methods and discuss potential avenues for improvement, especially in light of the inherent challenges and constraints posed by the fossil record.

- The methods section is great. But could you explain how the different variables in the MBD were defined and calculated? Are there any limitations to this data?

Thank you for your comment. We have expanded our explanations regarding the construction of these variables, provided additional details on how they were compiled, and discuss their limitations.

Reviewer #2 (Remarks to the Author):

With pleasure, I read this manuscript describing a complex issue like as diversification and extinction of Hemiptera. It's an excellent paper, unique and valuable. This paper is properly organised, the title is informative and concise, and the abstract is representative of the content. The authors prepare an introduction based on relevant references. Authors based on the huge material and modern methods during their analysis. Methods are well described, and all-sufficient methodological detail was that the experiments could be reproduced; all figures are necessary and well-labelled. The discussion and conclusion are supported by the results, and the results are clearly presented. All claims are convincing, and appropriately discussed in the context of previous literature. It is a huge article collecting information about the entire order of insects, discussing all groups of insects included in it and I'm sure that this article will clarify some previously controversial issues. It is a very important and helpful paper for the next researchers who will analyze a group of hemipteran species in the

future. I think, that the paper will be of interest to others in the field, not only for hemipterologists and maybe not only entomologists at all.

The manuscript is clearly written. I am not qualified to assess the quality of English in this paper, but the language is precise and understandable also for non-native speakers.

We greatly appreciate the reviewer's enthusiasm for our paper and share their views on its potential impact. At the same time, we acknowledge the limitations of our methodology, as detailed in the relevant section. Thank you for your valuable time and thoughtful review.

I have a few questions or suggestions and a lot of corrections in References.

Line 74. Hemiptera are characterized by a segmented rostrum with a multi-segmented sheet-like labium covering the mandibular and maxillary stylets (Emeljanov, 2002).

This sentence is not clear: only the labium is segmented; the rest parts of the mouthparts aren't segmented in Hemiptera. I suggest citing other papers, more modern (for example Brožek J. et al. 2019, 2020, 2022), not only Emeljanov, 2002.

Thank you for your comment. We completely agree that only the labium is segmented (ancestrally) in Hemiptera. We corrected our paragraph accordingly.

We added the following references:

Cobben, R.H. (1978). *Evolutionary Trends in Heteroptera — Part 2: Mouthpart-Structures and Feeding Strategies*. Landbouwhogeschool Wageningen, Netherlands, 78–5, voor Entomologie Mededelingen, 289: 1–407.

Taszakowski, A., Masłowski, A. and Brožek, J. (2023) Labial Sensory Organs of Two Leptoglossus Species (Hemiptera: Coreidae): Their Morphology and Supposed Function. *Insects*, 14(1), 30. <https://doi.org/10.3390/insects14010030>

Brožek, J., Mróz, E., Wylężek, D., Depa, Ł. and Węgierek, P. (2015) The structure of extremely long mouthparts in the aphid genus *Stomaphis* Walker (Hemiptera: Sternorrhyncha: Aphididae). *Zoomorphology* 134, 431–445. <https://doi.org/10.1007/s00435-015-0266-7>

Line 81. What taxonomic system use authors in this paper? Which one? After whom?

Thank you for your question. For this study, we adhered to the taxonomic and systematic framework for Hemiptera proposed by Szwedo (2018), which remains the most practical approach for integrating both extant and extinct hemipteran lineages. However, this classification does not explicitly recognize Auchenorrhyncha as a formal rank. To address this, we treated Auchenorrhyncha as a monophyletic group, a decision thoroughly discussed in the “*Systematic Framework*” subsection of the **Materials and Methods** section.

Reference:

Szwedo, J. (2018) The unity, diversity and conformity of bugs (Hemiptera) through time. *Earth and Environmental Science Transactions of the Royal Society of Edinburgh*, 107, 109–128. <https://doi.org/10.1017/S175569101700038X>

Lines 87-93. Heteroptera switched to phytophagy several times, but each time differently; it was not just a return to phloem (in some, yes) but to fruits, seeds, algae, fungi, etc. (Panizzi et al. 2021).

Thank you for highlighting this important point. We have revised the relevant section to address your remark and incorporated the reference you provided to enhance the discussion.

Line 106. The oldest Nepomorpha are Triassic, so they did not re-enter the water, but settled freshwater during the Jurassic Lacustrine revolution, similarly Gerromorpha probably then became widespread.

Thank you for your comment. There is ongoing debate regarding the deep relationships within Heteroptera phylogeny. Some studies propose that Nepomorpha represents the sister lineage to all

other heteropteran groups (Li et al., 2012; Weirauch et al., 2019). This hypothesis aligns with the fossil record, as the earliest known true bug representatives belong to Nepomorpha and are recorded from the Middle Triassic of France (Shcherbakov, 2010). However, recent molecular analyses, particularly those utilizing phylogenomic data (Wang et al., 2016; Johnson et al., 2018), suggest an alternative topology. These studies position Nepomorpha as the sister lineage to (Leptopodomorpha, (Pentatomorpha, Cimicomorpha)) within a clade known as Panheteroptera, which itself is the sister lineage to ((Enicocephalomorpha, Dipsocoromorpha) Gerromorpha).

Regarding ancestral ecological conditions, terrestrial habitats are likely the ancestral state for Heteroptera. Aquatic and semi-aquatic true bugs, such as those within Nepomorpha and Gerromorpha, are thought to have independently invaded aquatic environments on at least three occasions (Weirauch et al., 2019). Evidence from the Middle Triassic suggests that terrestrial Heteroptera were already present at this time, further supporting the idea of independent ecological shifts (Montagna et al., 2019). Consequently, the rise and diversification of both Nepomorpha and Gerromorpha are more plausibly associated with distinct and independent adaptations to aquatic lifestyles.

Given the complexity of these relationships and to streamline the narrative flow of our introduction, we have decided to omit this concise yet challenging-to-integrate paragraph, as noted in our response to Reviewer 1's comments. Thank you for your understanding and valuable feedback.

References:

- Johnson, K.P., Dietrich, C.H., Friedrich, F., Beutel, R.G., Wipfler, B., Peters, R.S., Allen, J.M., Petersen, M., Donath, A., Walden, K.K.O., Kozlov, A.M., Podsiadlowski, L., Mayer, C., Meusemann, K., Vasilikopoulos, A., Waterhouse, R.M., Cameron, S.L., Weirauch, C., Swanson, D.R., Percy, D.M., Hardy, N.B., Terry, I., Liu, S., Zhou, X., Misof, B., Robertson, H.M. and Yoshizawa, K. (2018) Phylogenomics and the evolution of hemipteroid insects. *Proceedings of the National Academy of Sciences of the United States of America*, 115 (50): 12775–12780. <https://doi.org/10.1073/pnas.1815820115>
- Li, M., Tian, Y., Zhao, Y. and Bu, W. (2012) Higher Level Phylogeny and the First Divergence Time Estimation of Heteroptera (Insecta: Hemiptera) Based on Multiple Genes. *PLoS One*, 7(2): e32152. <https://doi.org/10.1371/journal.pone.0032152>
- Montagna, M., Tong, K. J., Magoga, G., Strada, L., Tintori, A., Ho, S.Y.W. and Lo, N. (2019) Recalibration of the insect evolutionary time scale using Monte San Giorgio fossils suggests survival of key lineages through the End-Permian Extinction. *Proceedings of the Royal Society B*, 286: 20191854. <https://doi.org/10.1098/rspb.2019.1854>
- Shcherbakov, D.E. (2010) The earliest true bugs and aphids from the Middle Triassic of France (Hemiptera). *Russian Entomological Journal*, 19: 179–182. <http://dx.doi.org/10.15298/rusentj.19.3.04>
- Wang, Y., Cui, Y., Rédei, D., Baňář, P., Xie, Q., Štys, P., Damgaard, J., Chen, P., Yi, W., Wang, Y., Dang, K., Li, C. and Bu W. (2016) Phylogenetic divergences of the true bugs (Insecta: Hemiptera: Heteroptera), with emphasis on the aquatic lineages: the last piece of the aquatic insect jigsaw originated in the Late Permian/Early Triassic. *Cladistics*, 32(4): 335–478. <https://doi.org/10.1111/cla.12137>
- Weirauch C., Schuh, R.T., Cassis, G. and Wheeler, W.C. (2019) Revisiting habitat and lifestyle transitions in Heteroptera (Insecta: Hemiptera): insights from a combined morphological and molecular phylogeny. *Cladistics*, 35(1): 67–105. <https://doi.org/10.1111/cla.12233>

Line 118. During the Oligocene and Miocene, there were mainly open areas - herbs, then there was a change in metabolism from C4 to C3 in grasses hence the shift and transition in some to other endosymbionts, hence the radiation of Delphacidae and Cicadellidae. Also in some of the aphids that like colder climates and also some of the maggots associated with grasses.

Thank you for your insightful remark. We have incorporated these pertinent observations to emphasize not only the late radiation of Cicadellidae and Membracidae (Cicadomorpha) but also the diversification events observed in certain aphids and planthoppers (Delphacidae).

References corrections:

Papers not cited in the text: Alroy, J. (2016); Barnosky, A.D. (1999).; Barnosky, A.D. (1999); Bernardi, M., Gianolla, P., Petti, F.M., Mietto, P. and Benton, M.J. (2018); Condamine, F.L., Rolland, J. and Morlon, H. (2019); Depa, L., Kaszyca-Taszakowska, N., Taszakowski, A. and Kanturski, M. 2020; Foote, M. and Miller I.A. (2007); Lovegrove, B.G. and Mowoe, M.O. (2015); Nel, A. (1997); Popov, Y.A. (1968); Van Valen, L. 1973.

Thank you for pointing out these oversights. We have reviewed and removed any references that were not cited in the manuscript to ensure consistency and clarity.

Lack of references: Louca and Pennell, 2020; Li et al., 2017; Labandeira and Li, 2021; Burckhardt et al., 2022; Labandeira, 1997; Prokop et al., 2019; Rainford et al., 2014.

Thank you for identifying these omissions and discrepancies. We would like to clarify that some differences stem from variations in publication dates, particularly between the online release of articles and their final inclusion in a volume, which can often take one to two years:

1. The full reference for Louca and Pennell (2020) was already included in the manuscript but lacked the publication date, which we have now added.
2. The citation for Burckhardt et al. was corrected by updating the year from 2022 to 2023.
3. The citation for Prokop et al. (2019) was removed and replaced with the following reference: Shear, W.A., & Kukulová-Peck, J. (1990). The ecology of Paleozoic terrestrial arthropods: the fossil evidence. *Canadian Journal of Zoology*, 68(8), 1807–1834.

Different years in the text and references: Pyron and Burbrink, 2013; Jiang et al., 2022; Jouault et al., 2024 a or b? Wrong order, not alphabetical, for example, Cryan/ Condamine.

Thank you for highlighting these inconsistencies. We have made the necessary corrections in the manuscript.

In my opinion, the overall quality of the work is high and I warmly recommend this article for publication in *Communications Biology* after small corrections.

We would like to take this opportunity to once again thank both reviewers for their time and valuable feedback. We believe that the revised version of the manuscript has significantly improved as a result of their insightful comments.